# Globally Optimal Training of Neural Networks with Threshold Activation Functions

**Tolga Ergen, Halil Ibrahim Gulluk, Jonathan Lacotte & Mert Pilanci**
Department of Electrical Engineering
Stanford University
Stanford, CA 94305, USA
{ergen,gulluk,lacotte,pilanci}@stanford.edu

## Abstract

Threshold activation functions are highly preferable in neural networks due to their efficiency in hardware implementations. Moreover, their mode of operation is more interpretable and resembles that of biological neurons. However, traditional gradient based algorithms such as Gradient Descent cannot be used to train the parameters of neural networks with threshold activations since the activation function has zero gradient except at a single non-differentiable point. To this end, we study weight decay regularized training problems of deep neural networks with threshold activations. We first show that regularized deep threshold network training problems can be equivalently formulated as a standard convex optimization problem, which parallels the LASSO method, provided that the last hidden layer width exceeds a certain threshold. We also derive a simplified convex optimization formulation when the dataset can be shattered at a certain layer of the network. We corroborate our theoretical results with various numerical experiments.

## 1 Introduction

In the past decade, deep neural networks have proven remarkably useful in solving challenging problems and become popular in many applications. The choice of activation plays a crucial role in their performance and practical implementation. In particular, even though neural networks with popular activation functions such as ReLU are successfully employed, they require advanced computational resources in training and evaluation, e.g., Graphical Processing Units (GPUs) (Coates et al., 2013). Consequently, training such deep networks is challenging especially without sophisticated hardware. On the other hand, the threshold activation offers a multitude of advantages: (1) computational efficiency, (2) compression/quantization to binary latent dimension, (3) interpretability. Unfortunately, gradient based optimization methods fail in optimizing threshold activation networks due to the fact that the gradient is zero almost everywhere. To close this gap, we analyze the training problem of deep neural networks with the threshold activation function defined as

$$\sigma_s(x) := s\mathbb{1}\{x \geq 0\} = \begin{cases} s & \text{if } x \geq 0 \\ 0 & \text{otherwise} \end{cases}, \tag{1}$$

where $s \in \mathbb{R}$ is a trainable amplitude parameter for the neuron. Our main result is that globally optimal deep threshold networks can be trained by solving a convex optimization problem.

### 1.1 Why should we care about threshold networks?

Neural networks with threshold activations are highly desirable due to the following reasons:

- Since the threshold activation (1) is restricted to take values in $\{0, s\}$, threshold neural network models are far more suitable for hardware implementations (Bartlett & Downs, 1992; Corwin et al., 1994). Specifically, these networks have significantly lower memory footprint, less computational complexity, and consume less energy (Helwegen et al., 2019).
- Modern neural networks have extremely large number of full precision trainable parameters so that several computational barriers emerge during hardware implementations. One approach to

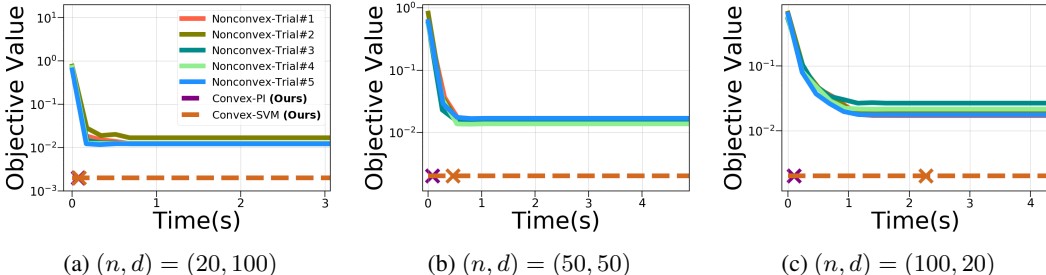

(a) $(n, d) = (20, 100)$  (b) $(n, d) = (50, 50)$  (c) $(n, d) = (100, 20)$

Figure 1: Training comparison of our convex program in (7) with the non-convex training heuristic STE. We also indicate the time taken to solve the convex programs with markers. For non-convex STE, we repeat the training with 5 different initializations. In each case, our convex training algorithms achieve lower objective than all the non-convex heuristics (see Appendix B.5 for details).

mitigate these issues is reducing the network size by grouping the parameters via a hash function (Hubara et al., 2017; Chen et al., 2015). However, this still requires full precision training before the application of the hash function and thus fails to remedy the computational issues. On the other hand, neural networks with threshold activations need a minimal amount of bits.

- Another approach to reduce the complexity is to quantize the weights and activations of the network (Hubara et al., 2017) and the threshold activation is inherently in a two level quantized form.
- The threshold activation is a valid model to simulate the behaviour of a biological neuron as detailed in Jain et al. (1996). Therefore, progress in this research field could shed light into the connection between biological and artificial neural networks.

## 1.2  RELATED WORK

Although threshold networks are essential for several practical applications as detailed in the previous section, training their parameters is a difficult non-differentiable optimization problem due to the discrete nature in (1). For training of deep neural networks with popular activations, the common practice is to use first order gradient based algorithms such as Gradient Descent (GD) since the well known backpropagation algorithm efficiently calculates the gradient with respect to parameters. However, the threshold activation in (1) has zero gradient except at a single non-differentiable point zero, and therefore, one cannot directly use gradient based algorithms to train the parameters of the network. In order to remedy this issue numerous heuristic algorithms have been proposed in the literature as detailed below but they still fail to globally optimize the training objective (see Figure 1).

The Straight-Through Estimator (STE) is a widely used heuristics to train threshold networks (Bengio et al., 2013; Hinton, 2012). Since the gradient is zero almost everywhere, Bengio et al. (2013); Hinton (2012) proposed replacing the threshold activation with the identity function during only the backward pass. Later on, this approach is extended to employ various forms of the ReLU activation function, e.g., clipped ReLU, vanilla ReLU, Leaky ReLU (Yin et al., 2019b; Cai et al., 2017; Xiao et al.), during the backward pass. Additionally, clipped versions of the identity function were also used as an alternative to STE (Hubara et al., 2017; Courbariaux et al., 2016; Rastegari et al., 2016).

## 1.3  CONTRIBUTIONS

- We introduce polynomial-time trainable convex formulations of regularized deep threshold network training problems provided that a layer width exceeds a threshold detailed in Table 1.
- In Theorem 2.2, we prove that the original non-convex training problem for two-layer networks is equivalent to standard convex optimization problems.
- We show that deep threshold network training problems are equivalent to standard convex optimization problems in Theorem 3.2. In stark contrast to two-layer networks, deep threshold networks can have a richer set of hyperplane arrangements due to multiple nonlinear layers (see Lemma 3.5).
- In Section 3.1, we characterize the evolution of the set of hyperplane arrangements and consequently hidden layer representation space as a recursive process (see Figure 3) as the network gets deeper.
- We prove that when a certain layer width exceeds $\mathcal{O}(\sqrt{n}/L)$, the regularized $L$-layer threshold network training further simplifies to a problem that can be solved in $\mathcal{O}(n)$ time.

Table 1: Summary of our results for the optimization of weight decay regularized threshold network training problems ($n$: # of data samples, $d$: feature dimension, $m_l$: # of hidden neurons in layer $l$, $r$: rank of the training data matrix, $m^*$: critical width, i.e., # of neurons that obeys $0 \leq m^* \leq n + 1$)

| Result | Depth | Complexity | Minimum width | Globally optimal |
|---|---|---|---|---|
| Theorem 2.2 | 2 | $\mathcal{O}(n^{3r})$ | $m \geq m^*$ | ✓(convex opt) |
| Theorem 2.3 | 2 | $\mathcal{O}(n)$ | $m \geq n + 2$ | ✓(convex opt) |
| Theorem 3.2 | $L$ | $\mathcal{O}(n^{3r} \prod_{l=1}^{L-2} m_l)$ | $m_{L-1} \geq m^*$ | ✓(convex opt) |
| Corollary 3.4 | $L$ | $\mathcal{O}(n)$ | $\exists l : m_l \geq \mathcal{O}(\sqrt{n}/L)$ | ✓(convex opt) |

**Notation and preliminaries:** We use lowercase and uppercase bold letters to denote vectors and matrices, respectively. We also use $[n]$ to denote the set $\{1, 2, \ldots, n\}$. We denote the training data matrix consisting of $d$ dimensional $n$ samples as $\mathbf{X} \in \mathbb{R}^{n \times d}$, label vector as $\mathbf{y} \in \mathbb{R}^n$, and the $l^{th}$ layer weights of a neural network as $\mathbf{W}^{(l)} \in \mathbb{R}^{m_{l-1} \times m_l}$, where $m_0 = d, m_L = 1$.

## 2 TWO-LAYER THRESHOLD NETWORKS

We first consider the following two-layer threshold network

$$f_{\theta,2}(\mathbf{X}) = \sigma_{\mathbf{s}}(\mathbf{X}\mathbf{W}^{(1)})\mathbf{w}^{(2)} = \sum_{j=1}^{m} s_j \mathbb{1}\{\mathbf{X}\mathbf{w}_j^{(1)} \geq 0\} w_j^{(2)}, \tag{2}$$

where the set of the trainable parameters are $\mathbf{W}^{(1)} \in \mathbb{R}^{d \times m}, \mathbf{s} \in \mathbb{R}^m, \mathbf{w}^{(2)} \in \mathbb{R}^m$ and $\theta$ is a compact representation for the parameters, i.e., $\theta := \{\mathbf{W}^{(1)}, \mathbf{s}, \mathbf{w}^{(2)}\}$. Note that we include bias terms by concatenating a vector of ones to $\mathbf{X}$. Next, consider the weight decay regularized training objective

$$\mathcal{P}_2^{\text{noncvx}} := \min_{\mathbf{W}^{(1)}, \mathbf{s}, \mathbf{w}^{(2)}} \frac{1}{2} \|f_{\theta,2}(\mathbf{X}) - \mathbf{y}\|_2^2 + \frac{\beta}{2} \sum_{j=1}^{m} \left( \|\mathbf{w}_j^{(1)}\|_2^2 + |s_j|^2 + |w_j^{(2)}|^2 \right). \tag{3}$$

Now, we apply a scaling between variables $s_j$ and $w_j^{(2)}$ to reach an equivalent optimization problem.

**Lemma 2.1** (Optimal scaling). *The training problem in (3) can be equivalently stated as*

$$\mathcal{P}_2^{\text{noncvx}} = \min_{\theta \in \Theta_s} \frac{1}{2} \|f_{\theta,2}(\mathbf{X}) - \mathbf{y}\|_2^2 + \beta \|\mathbf{w}^{(2)}\|_1, \tag{4}$$

*where* $\Theta_s := \{\theta : |s_j| = 1, \forall j \in [m]\}$.

We next define the set of hyperplane arrangement patterns of the data matrix $\mathbf{X}$ as

$$\mathcal{H}(\mathbf{X}) := \{\mathbb{1}\{\mathbf{X}\mathbf{w} \geq 0\} : \mathbf{w} \in \mathbb{R}^d\} \subset \{0, 1\}^n. \tag{5}$$

We denote the distinct elements of the set $\mathcal{H}(\mathbf{X})$ by $\mathbf{d}_1, \ldots, \mathbf{d}_P \in \{0, 1\}^n$, where $P := |\mathcal{H}(\mathbf{X})|$ is the number of hyperplane arrangements. Using this fixed set of hyperplane arrangements $\{\mathbf{d}_i\}_{i=1}^{P}$, we next prove that (4) is equivalent to the standard Lasso method (Tibshirani, 1996).

**Theorem 2.2.** *Let* $m \geq m^*$, *then the non-convex regularized training problem (4) is equivalent to*

$$\mathcal{P}_2^{\text{cvx}} = \min_{\mathbf{w} \in \mathbb{R}^P} \frac{1}{2} \|\mathbf{D}\mathbf{w} - \mathbf{y}\|_2^2 + \beta \|\mathbf{w}\|_1, \tag{6}$$

*where* $\mathbf{D} = [\mathbf{d}_1 \quad \mathbf{d}_2 \quad \ldots \quad \mathbf{d}_P]$ *is a fixed* $n \times P$ *matrix. Here,* $m^*$ *is the cardinality of the optimal solution, which satisfies* $m^* \leq n + 1$. *Also, it holds that* $\mathcal{P}_2^{\text{noncvx}} = \mathcal{P}_2^{\text{cvx}}$.

Theorem 2.2 proves that the original non-convex training problem in (3) can be equivalently solved as a standard convex Lasso problem using the hyperplane arrangement patterns as features. Surprisingly, the non-zero support of the convex optimizer in (6) matches to that of the optimal weight-decay regularized threshold activation neurons in the non-convex problem (3). This brings us two major advantages over the standard non-convex training:

- Since (6) is a standard convex problem, it can be globally optimized without resorting to non-convex optimization heuristics, e.g., initialization schemes, learning rate schedules etc.
- Since (6) is a convex Lasso problem, there exists many efficient solvers (Efron et al., 2004).

## 2.1 SIMPLIFIED CONVEX FORMULATION FOR COMPLETE ARRANGEMENTS

We now show that if the set of hyperplane arrangements of $\mathbf{X}$ is complete, i.e., $\mathcal{H} = \{0,1\}^n$ contains all boolean sequences of length $n$, then the non-convex optimization problem in (3) can be simplified. We call these instances *complete arrangements*. In the case of two-layer threshold networks, complete arrangements emerge when the width of the network exceeds a threshold, specifically $m \geq n + 2$.

We note that the $m \geq n$ regime, also known as memorization, has been extensively studied in the recent literature (Bubeck et al., 2020; de Dios & Bruna, 2020; Pilanci & Ergen, 2020; Rosset et al., 2007). Particularly, these studies showed that as long as the width exceeds the number of samples, there exists a neural network model that can exactly fit an arbitrary dataset. Vershynin (2020); Bartlett et al. (2019) further improved the condition on the width by utilizing the expressive power of deeper networks and developed more sophisticated weight construction algorithms to fit the data.

**Theorem 2.3.** *We assume that the set of hyperplane arrangements of $\mathbf{X}$ is complete, i.e., equal to the set of all length-$n$ Boolean sequences" $\mathcal{H} = \{0,1\}^n$. Suppose $m \geq n + 2$, then (4) is equivalent to*

$$\mathcal{P}_{v2}^{\mathrm{cvx}} := \min_{\boldsymbol{\delta} \in \mathbb{R}^n} \frac{1}{2} \|\boldsymbol{\delta} - \mathbf{y}\|_2^2 + \beta(\|(\boldsymbol{\delta})_+\|_\infty + \|(-\boldsymbol{\delta})_+\|_\infty). \tag{7}$$

*and it holds that $\mathcal{P}_2^{\mathrm{noncvx}} = \mathcal{P}_{v2}^{\mathrm{cvx}}$. Also, one can construct an optimal network with $n + 2$ neurons in time $\mathcal{O}(n)$ based on the optimal solution to the convex problem (7).*

Based on Theorem 2.3, when the data matrix can be shattered, i.e., all $2^n$ possible $\mathbf{y} \in \{0,1\}^n$ labelings of the data points can be separated via a linear classifier, it follows that the set of hyperplane arrangements is complete. Consequently, the non-convex problem in (3) further simplifies to (7).

## 2.2 TRAINING COMPLEXITY

We first briefly summarize our complexity results for solving (6) and (7), and then provide the details of the derivations below. Our analysis reveals two interesting regimes:

- (incomplete arrangements) When $n + 1 \geq m \geq m^*$, we can solve (6) in $\mathcal{O}(n^{3r})$ complexity, where $r := \mathrm{rank}(\mathbf{X})$. Notice this is **polynomial-time** whenever the rank $r$ is fixed.
- (complete arrangements) When $m \geq n + 2$, we can solve (7) in closed-form, and the reconstruction of the non-convex parameters $\mathbf{W}^{(1)}$ and $\mathbf{w}^{(2)}$ only $\mathcal{O}(n)$ time independent of $d$.

**Computational complexity of** (6)**:** To solve the optimization problem in (6), we first enumerate all possible hyperplane arrangements $\{\mathbf{d}_i\}_{i=1}^P$. It is well known that given a rank-$r$ data matrix the number of hyperplane arrangements $P$ is upper bounded by (Stanley et al., 2004; Cover, 1965)

$$P \leq 2 \sum_{k=0}^{r-1} \binom{n-1}{k} \leq 2r \left(\frac{e(n-1)}{r}\right)^r, \tag{8}$$

where $r = \mathrm{rank}(\mathbf{X}) \leq \min(n, d)$. Furthermore, these can be enumerated in $\mathcal{O}(n^r)$ (Edelsbrunner et al., 1986). Then, the complexity for solving (6) is $\mathcal{O}(P^3) \approx \mathcal{O}(n^{3r})$ (Efron et al., 2004).

**Computational complexity of** (7)**:** The problem in (7) is the proximal operator of a polyhedral norm. Since the problem is separable over the positive and negative parts of the parameter vector $\boldsymbol{\delta}$, the optimal solution can be obtained by applying two proximal steps (Parikh & Boyd, 2014). As noted in Theorem 2.3, the reconstruction of the non-convex parameters $\mathbf{W}^{(1)}$ and $\mathbf{w}^{(2)}$ requires $\mathcal{O}(n)$ time.

## 2.3 A GEOMETRIC INTERPRETATION

To provide a geometric interpretation, we consider the weakly regularized case where $\beta \to 0$. In this case, (6) reduces to the following minimum norm interpolation problem

$$\min_{\mathbf{w} \in \mathbb{R}^P} \|\mathbf{w}\|_1 \quad \text{s.t.} \quad \mathbf{D}\mathbf{w} = \mathbf{y}. \tag{9}$$

**Proposition 2.4.** *The minimum $\ell_1$ norm interpolation problem in (9) can be equivalently stated as*

$$\min_{t \geq 0} t \quad \text{s.t.} \quad \mathbf{y} \in t\mathrm{Conv}\{\pm \mathbf{d}_j, \forall j \in [P]\},$$

where $\mathrm{Conv}(\mathcal{A})$ denotes the convex hull of a set $\mathcal{A}$. This corresponds to the gauge function (see Rockafellar (2015)) of the hyperplane arrangement patterns and their negatives. We provide visualizations of the convex set $\mathrm{Conv}\{\pm\mathbf{d}_j, \forall j \in [P]\}$ in Figures 3 and 4 (see Appendix) for Example 3.1.

Proposition 2.4 implies that the non-convex threshold network training problem in (3) implicitly represents the label vector $\mathbf{y}$ as the convex combination of the hyperplane arrangements determined by the data matrix $\mathbf{X}$. Therefore, we explicitly characterize the representation space of threshold networks. We also remark that this interpretation extends to arbitrary depth as shown in Theorem 3.2.

## 2.4 HOW TO OPTIMIZE THE HIDDEN LAYER WEIGHTS?

After training via the proposed convex programs in (6) and (7), we need to reconstruct the layer weights of the non-convex model in (2). We first construct the optimal hyperplane arrangements $\mathbf{d}_i$ for (7) as detailed in Appendix A.4. Then, we have the following prediction model (2). Notice changing $\mathbf{w}_j^{(1)}$ to any other vector $\mathbf{w}_j'$ with same norm and such that $\mathbb{1}\{\mathbf{X}\mathbf{w}_j^{(1)} \geq 0\} = \mathbb{1}\{\mathbf{X}\mathbf{w}_j' \geq 0\}$ does not change the optimal training objective. Therefore, there are multiple global optima and the one we choose might impact the generalization performance as discussed in Section 5.

## 3 DEEP THRESHOLD NETWORKS

We now analyze $L$-layer parallel deep threshold networks model with $m_{L-1}$ subnetworks defined as

$$f_{\theta,L}(\mathbf{X}) = \sum_{k=1}^{m_{L-1}} \sigma_{\mathbf{s}^{(L-1)}}(\mathbf{X}_k^{(L-2)}\mathbf{w}_k^{(L-1)})w_k^{(L)}, \tag{10}$$

where $\theta := \{\{\mathbf{W}_k^{(l)}, \mathbf{s}_k^{(l)}\}_{l=1}^L\}_{k=1}^{m_{L-1}}, \theta \in \Theta := \{\theta : \mathbf{W}_k^{(l)} \in \mathbb{R}^{m_{l-1} \times m_l}, \mathbf{s}_k^{(l)} \in \mathbb{R}^{m_l}, \forall l, k\}$,

$$\mathbf{X}_k^{(0)} := \mathbf{X}, \qquad\qquad \mathbf{X}_k^{(l)} := \sigma_{\mathbf{s}_k^{(l)}}(\mathbf{X}_k^{(l-1)}\mathbf{W}_k^{(l)}), \forall l \in [L-1],$$

and the subscript $k$ is the index for the subnetworks (see Ergen & Pilanci (2021a;b); Wang et al. (2023) for more details about parallel networks). We next show that the standard weight decay regularized training problem can be cast as an $\ell_1$-norm minimization problem as in Lemma 2.1.

**Lemma 3.1.** *The following $L$-layer regularized threshold network training problem*

$$\mathcal{P}_L^{\mathrm{noncvx}} = \min_{\theta \in \Theta} \frac{1}{2}\|f_{\theta,L}(\mathbf{X}) - \mathbf{y}\|_2^2 + \frac{\beta}{2}\sum_{k=1}^{m_{L-1}}\sum_{l=1}^{L}(\|\mathbf{W}_k^{(l)}\|_F^2 + \|\mathbf{s}_k^{(l)}\|_2^2) \tag{11}$$

*can be reformulated as*

$$\mathcal{P}_L^{\mathrm{noncvx}} = \min_{\theta \in \Theta_s} \frac{1}{2}\|f_{\theta,L}(\mathbf{X}) - \mathbf{y}\|_2^2 + \beta\|\mathbf{w}^{(L)}\|_1, \tag{12}$$

*where $\Theta_s := \{\theta \in \Theta : |s_k^{(L-1)}| = 1\}$.*

## 3.1 CHARACTERIZING THE SET OF HYPERPLANE ARRANGEMENTS FOR DEEP NETWORKS

We first define hyperplane arrangements for $L$-layer networks with a single subnetwork (i.e., $m_{L-1} = 1$, thus, we drop the index $k$). We denote the set of hyperplane arrangements as

$$\mathcal{H}_L(\mathbf{X}) := \{\mathbb{1}\{\mathbf{X}^{(L-2)}\mathbf{w}^{(L-1)} \geq 0\} : \theta \in \Theta\}.$$

We also denote the elements of the set $\mathcal{H}_L(\mathbf{X})$ by $\mathbf{d}_1, \ldots, \mathbf{d}_{P_{L-1}} \in \{0,1\}^n$ and $P_{L-1} = |\mathcal{H}_L(\mathbf{X})|$ is the number of hyperplane arrangements in the layer $L-1$.

To construct the hyperplane arrangement matrix $\mathbf{D}^{(l)}$, we define a matrix valued operator as follows

$$\mathbf{D}^{(1)} := \mathcal{A}(\mathbf{X}) = [\mathbf{d}_1 \quad \mathbf{d}_2 \quad \ldots \quad \mathbf{d}_{P_1}] , \quad \mathbf{D}^{(l+1)} := \bigsqcup_{|\mathcal{S}|=m_l} \mathcal{A}\left(\mathbf{D}_{\mathcal{S}}^{(l)}\right), \forall l \in [L-2]. \tag{13}$$

Here, the operator $\mathcal{A}(\cdot)$ outputs a matrix whose columns contain all possible hyperplane arrangements corresponding to its input matrix as in (5). In particular, $\mathbf{D}^{(1)}$ denotes the arrangements for the first layer given the input matrix $\mathbf{X}$. The notation $\mathbf{D}_{\mathcal{S}}^{(l)} \in \{0,1\}^{n \times m_l}$ denotes the submatrix of $\mathbf{D}^{(l)} \in \{0,1\}^{n \times P_l}$ indexed by the subset $\mathcal{S}$ of its columns, where the index $\mathcal{S}$ runs over all subsets of size $m_l$. Finally, $\sqcup$ is an operator that takes a union of these column vectors and outputs a matrix of size $n \times P_{l+1}$ containing these as columns. Note that we may omit repeated columns and denote the total number of unique columns as $P_{l+1}$, since this does not change the value of our convex program.

We next provide an analytical example describing the construction of the matrix $\mathbf{D}^{(l)}$.

**Example 3.1.** *We illustrate an example with the training data $\mathbf{X} = [-1 \ 1; 0 \ 1; 1 \ 1] \in \mathbb{R}^{3 \times 2}$. Inspecting the data samples (rows of $\mathbf{X}$), we observe that all possible arrangement patterns are*

$$\mathbf{D}^{(1)} = \mathcal{A}(\mathbf{X}) = \begin{bmatrix} 0 & 0 & 0 & 1 & 1 & 1 \\ 0 & 0 & 1 & 1 & 1 & 0 \\ 0 & 1 & 1 & 1 & 0 & 0 \end{bmatrix} \implies P_1 = 6. \tag{14}$$

*For the second layer, we first specify the number of neurons in the first layer as $m_1 = 2$. Thus, we need to consider all possible column pairs in (14). We have*

$$\mathbf{D}^{(1)}_{\{1,2\}} = \begin{bmatrix} 0 & 0 \\ 0 & 0 \\ 0 & 1 \end{bmatrix} \implies \mathcal{A}\left(\begin{bmatrix} 0 & 0 \\ 0 & 0 \\ 0 & 1 \end{bmatrix}\right) = \begin{bmatrix} 0 & 0 & 1 & 1 \\ 0 & 0 & 1 & 1 \\ 0 & 1 & 1 & 0 \end{bmatrix}$$

$$\mathbf{D}^{(1)}_{\{1,3\}} = \begin{bmatrix} 0 & 0 \\ 0 & 1 \\ 0 & 1 \end{bmatrix} \implies \mathcal{A}\left(\begin{bmatrix} 0 & 0 \\ 0 & 1 \\ 0 & 1 \end{bmatrix}\right) = \begin{bmatrix} 0 & 0 & 1 & 1 \\ 0 & 1 & 1 & 0 \\ 0 & 1 & 1 & 0 \end{bmatrix}$$

$$\vdots$$

*We then construct the hyperplane arrangement matrix as*

$$\mathbf{D}^{(2)} = \bigsqcup_{|\mathcal{S}|=2} \mathcal{A}\left(\mathbf{D}_{\mathcal{S}}^{(1)}\right) = \begin{bmatrix} 0 & 0 & 0 & 0 & 1 & 1 & 1 & 1 \\ 0 & 0 & 1 & 1 & 0 & 0 & 1 & 1 \\ 0 & 1 & 0 & 1 & 0 & 1 & 0 & 1 \end{bmatrix},$$

*which shows that $P_2 = 8$. Consequently, we obtain the maximum possible arrangement patterns, i.e., $\{0,1\}^3$, in the second layer even though we are not able to obtain some of these patterns in the first layer in (14). We also provide a three dimensional visualization of this example in Figure 3.*

## 3.2 POLYNOMIAL-TIME TRAINABLE CONVEX FORMULATION

Based on the procedure described in Section 3.1 to compute the arrangement matrix $\mathbf{D}^{(l)}$, we now derive an exact formulation for the non-convex training problem in (12).

**Theorem 3.2.** *Suppose that $m_{L-1} \geq m^*$, then the non-convex training problem (12) is equivalent to*

$$\mathcal{P}_L^{\mathrm{cvx}} = \min_{\mathbf{w} \in \mathbb{R}^{P_{L-1}}} \frac{1}{2} \left\| \mathbf{D}^{(L-1)}\mathbf{w} - \mathbf{y} \right\|_2^2 + \beta \|\mathbf{w}\|_1, \tag{15}$$

*where $\mathbf{D}^{(L-1)} \in \{0,1\}^{n \times P_{L-1}}$ is a fixed matrix constructed via (13) and $m^*$ denotes the cardinality of the optimal solution, which satisfies where $m^* \leq n + 1$. Also, it holds that $\mathcal{P}_L^{\mathrm{noncvx}} = \mathcal{P}_L^{\mathrm{cvx}}$.*

Theorem 3.2 shows that two-layer and deep networks simplify to very similar convex Lasso problems (i.e. (6) and (15)). However, the set of hyperplane arrangements is larger for deep networks as analyzed in Section 3.1. Thus, the structure of the diagonal matrix $\mathbf{D}$ and the problem dimensionality are significantly different for these problems.

## 3.3 SIMPLIFIED CONVEX FORMULATION

Here, we show that the data $\mathbf{X}$ can be shattered at a certain layer, i.e., $\mathcal{H}_l(\mathbf{X}) = \{0,1\}^n$ for a certain $l \in [L]$, if the number of hidden neurons in a certain layer $m_l$ satisfies $m_l \geq C\sqrt{n}/L$. Then we can alternatively formulate (11) as a simpler convex problem. Therefore, compared to the two-layer networks in Section 2.1, we substantially improve the condition on layer width by benefiting from the depth $L$, which also confirms the benign impact of the depth on the optimization.

**Lemma 3.3.** *If $\exists l, C$ such that $m_l \geq C\sqrt{n}/L$, then the set of hyperplane arrangements is complete, i.e., $\mathcal{H}_L(\mathbf{X}) = \{0,1\}^n$.*

We next use Lemma 3.3 to derive a simpler form of (11).

**Corollary 3.4.** *As a direct consequence of Theorem 2.3 and Lemma 3.3, the non-convex deep threshold network training problem in (12) can be cast as the following convex program*

$$\mathcal{P}_L^{\text{noncvx}} = \mathcal{P}_{v2}^{\text{cvx}} = \min_{\boldsymbol{\delta} \in \mathbb{R}^n} \frac{1}{2}\|\boldsymbol{\delta} - \mathbf{y}\|_2^2 + \beta(\|(\boldsymbol{\delta})_+\|_\infty + \|(-\boldsymbol{\delta})_+\|_\infty).$$

Surprisingly, both two-layer and deep networks share the same convex formulation in this case. However, notice that two-layer networks require a condition on the data matrix in Theorem 2.3 whereas the result in Corollary 3.4 requires a milder condition on the layer widths.

### 3.4 TRAINING COMPLEXITY

Here, we first briefly summarize our complexity results for the convex training of deep networks. Based on the convex problems in (15) and Corollary 3.4, we have two regimes:

- When $n+1 \geq m_{L-1} \geq m^*$, we solve (15) with $\mathcal{O}(n^{3r}\prod_{k=1}^{L-2} m_k)$ complexity, where $r := \text{rank}(\mathbf{X})$. Note that this is **polynomial-time** when $r$ and the number of neurons in each layer $\{m_l\}_{l=1}^{L-2}$ are constants.
- When $\exists l, C : m_l \geq C\sqrt{n}/L$, we solve (7) in closed-form, and the reconstruction of the non-convex parameters requires $\mathcal{O}(n)$ time as proven in Appendix A.9.

**Computational complexity for (15):** We first need to obtain an upperbound on the problem dimensionality $P_{L-1}$, which is stated in the next result.

**Lemma 3.5.** *The cardinality of the hyperplane arrangement set for an $L$-layer network $\mathcal{H}_L(\mathbf{X})$ can be bounded as $|\mathcal{H}_L(\mathbf{X})| = P_{L-1} \lesssim \mathcal{O}(n^r \prod_{k=1}^{L-2} m_k)$, where $r = \text{rank}(\mathbf{X})$ and $m_l$ denotes the number of hidden neurons in the $l^{th}$ layer.*

Lemma 3.5 shows that the set of hyperplane arrangements gets significantly larger as the depth of the network increases. However, the cardinality of this set is still a polynomial term since that $r < \min\{n, d\}$ and $\{m_l\}_{l=1}^{L-2}$ are fixed constants.

To solve (15), we first enumerate all possible arrangements $\{\mathbf{d}_i\}_{i=1}^{P_{L-1}}$ to construct the matrix $\mathbf{D}^{(L-1)}$. Then, we solve a standard convex Lasso problem, which requires $\mathcal{O}(P_{L-1}^3)$ complexity (Efron et al., 2004). Thus, based on Lemma 3.5, the overall complexity is $\mathcal{O}(P_{L-2}^3) \approx \mathcal{O}(n^{3r}\prod_{k=1}^{L-2} m_k)$.

**Computational complexity for (7):** Since Corollary 3.4 yields (7), the complexity is $\mathcal{O}(n)$ time.

## 4 EXTENSIONS TO ARBITRARY LOSS FUNCTIONS

In the previous sections, we considered squared error as the loss function to give a clear description of our approach. However, all the derivations extend to arbitrary convex loss. Now, we consider the regularized training problem with a convex loss function $\mathcal{L}(\cdot, \mathbf{y})$, e.g., hinge loss, cross entropy,

$$\min_{\theta \in \Theta} \mathcal{L}(f_{\theta, L}(\mathbf{X}), \mathbf{y}) + \frac{\beta}{2} \sum_{k=1}^{m_{L-1}} \sum_{l=1}^{L} (\|\mathbf{W}_k^{(l)}\|_F^2 + \|\mathbf{s}_k^{(l)}\|_2^2). \tag{16}$$

Then, we have the following generic loss results.

**Corollary 4.1.** *Theorem 3.2 implies that when $m_{L-1} \geq m^*$, (16) can be equivalently stated as*

$$\mathcal{P}_L^{\text{cvx}} = \min_{\mathbf{w} \in \mathbb{R}^{P_{L-1}}} \mathcal{L}\left(\mathbf{D}^{(L-1)}\mathbf{w}, \mathbf{y}\right) + \beta\|\mathbf{w}\|_1. \tag{17}$$

*Alternatively when $\mathcal{H}_L(\mathbf{X}) = \{0,1\}^n$, based on Corollary 3.4, the equivalent convex problem is*

$$\min_{\boldsymbol{\delta} \in \mathbb{R}^n} \mathcal{L}(\boldsymbol{\delta}, \mathbf{y}) + \beta(\|(\boldsymbol{\delta})_+\|_\infty + \|(-\boldsymbol{\delta})_+\|_\infty). \tag{18}$$

Corollary 4.1 shows that (17) and (18) are equivalent to the non-convex training problem in (16). More importantly, they can be globally optimized via efficient convex optimization solvers.

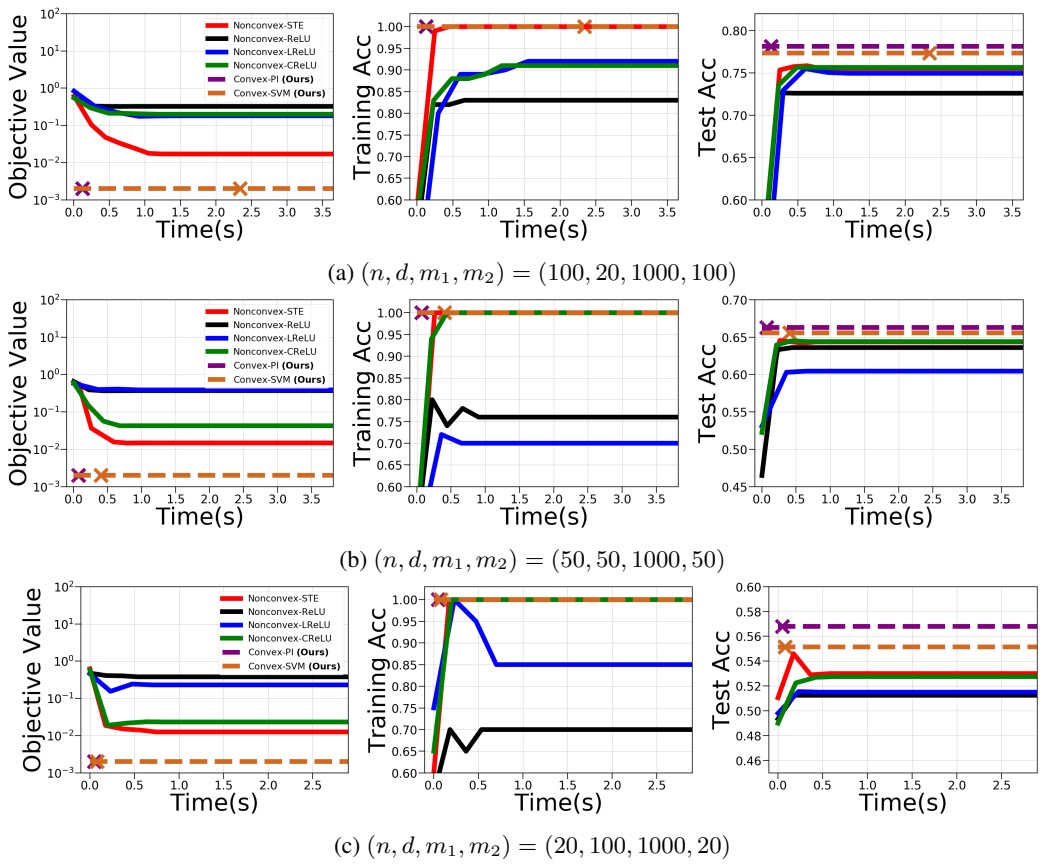

(a) $(n, d, m_1, m_2) = (100, 20, 1000, 100)$

(b) $(n, d, m_1, m_2) = (50, 50, 1000, 50)$

(c) $(n, d, m_1, m_2) = (20, 100, 1000, 20)$

Figure 2: In this figure, we compare the classification performance of three-layer threshold networks trained with the setup described in Figure 1 for a single initialization trial. This experiment shows that our convex training approach not only provides the globally optimal training performance but also generalize remarkably well on the test data (see Appendix B.5 for details).

## 5 EXPERIMENTS

In this section[1], we present numerical experiments verifying our theoretical results in the previous sections. As discussed in Section 2.4, after solving the proposed convex problems in (15) and (7), there exist multiple set of weight matrices yielding the same optimal objective value. Therefore, to have a good generalization performance on test data, we use some heuristic methods for the construction of the non-convex parameters $\{\mathbf{W}^{(l)}\}_{l=1}^L$. Below, we provide details regarding the weight construction and review some baseline non-convex training methods.

**Convex-Lasso:** To solve the problem (15), we first approximate arrangement patterns of the data matrix $\mathbf{X}$ by generating i.i.d. Gaussian weights $\mathbf{G} \in \mathbb{R}^{d \times \tilde{P}}$ and subsample the arrangement patterns via $\mathbb{1}[\mathbf{XG} \geq 0]$. Then, we use $\mathbf{G}$ as the hidden layer weights to construct the network. We repeat this process for every layer. Notice that here we sample a fixed subset of arrangements instead of enumerating all possible $P$ arrangements. Thus, this approximately solves (15) by subsampling its decision variables, however, it still performs significantly better than standard non-convex training.

**Convex-PI:** After solving (7), to recover the hidden layer weights of (2), we solve $\mathbb{1}\{\mathbf{Xw}_j^{(1)} \geq 0\} = \mathbf{d}_j$ as $\mathbf{w}_j^{(1)} = \mathbf{X}^\dagger \mathbf{d}_j$, where $\mathbf{X}^\dagger$ denotes the pseudo-inverse of $\mathbf{X}$. The resulting weights $\mathbf{w}_j^{(1)}$ enforce the preactivations $\mathbf{Xw}_j^{(1)}$ to be zero or one. Thus, if an entry is slightly higher or less than zero due to precision issues during the pseudo-inverse, it might give wrong output after the threshold activation. To avoid such cases, we use $0.5$ threshold in the test phase, i.e., $\mathbb{1}\{\mathbf{X}_{\text{test}}\mathbf{w}_j^{(1)} \geq 0.5\}$.

**Convex-SVM:** Another approach to solve $\mathbb{1}\{\mathbf{Xw}_j^{(1)} \geq 0\} = \mathbf{d}_j$ is to use Support Vector Machines

---

[1] We provide additional experiments and details in Appendix B.

Table 2: Test performance comparison on CIFAR-10 (Krizhevsky et al., 2014), MNIST (LeCun), and UCI (Dua & Graff, 2017) datasets. We repeat simulations over multiple seeds with two-layer networks and compare the mean/std of the test accuracies of non-convex heuristics trained with SGD with our convex program in (6). Our convex approach achieves highest test accuracy for 9 of 13 datasets whereas the best non-convex heuristic achieves the highest test accuracy only for 4 datasets.

| Dataset | Convex-Lasso (Ours) | | Nonconvex-STE | | Nonconvex-ReLU | | Nonconvex-LReLU | | Nonconvex-CReLU | |
|---|---|---|---|---|---|---|---|---|---|---|
| | Accuracy | Time(s) | Accuracy | Time(s) | Accuracy | Time(s) | Accuracy | Time(s) | Accuracy | Time(s) |
| CIFAR-10 | $0.816 \pm 0.008$ | $8.9 \pm 0.3$ | $0.81 \pm 0.004$ | $83.5 \pm 4.9$ | $0.803 \pm 0.004$ | $85.8 \pm 4.7$ | $0.798 \pm 0.004$ | $92.1 \pm 4.9$ | $0.808 \pm 0.004$ | $87.1 \pm 4.5$ |
| MNIST | $0.9991 \pm 1.9e^{-4}$ | $39.4 \pm 0.1$ | $0.9986 \pm 3.5e^{-4}$ | $61.3 \pm 0.04$ | $0.9984 \pm 4.6e^{-4}$ | $63.4 \pm 0.1$ | $0.9985 \pm 2.9e^{-4}$ | $75.5 \pm 0.1$ | $0.9985 \pm 2.9e^{-4}$ | $64.9 \pm 0.07$ |
| bank | $0.895 \pm 0.007$ | $7.72 \pm 1.02$ | $0.892 \pm 0.008$ | $5.83 \pm 0.06$ | $0.900 \pm 0.008$ | $5.96 \pm 0.12$ | $0.899 \pm 0.008$ | $8.41 \pm 0.12$ | $0.897 \pm 0.008$ | $6.35 \pm 0.11$ |
| chess-krvkp | $0.945 \pm 0.005$ | $5.34 \pm 0.38$ | $0.937 \pm 0.008$ | $6.78 \pm 0.20$ | $0.934 \pm 0.007$ | $6.17 \pm 0.21$ | $0.945 \pm 0.007$ | $7.44 \pm 0.10$ | $0.941 \pm 0.013$ | $6.15 \pm 0.47$ |
| mammographic | $0.818 \pm 0.014$ | $2.64 \pm 0.52$ | $0.808 \pm 0.011$ | $5.40 \pm 0.63$ | $0.803 \pm 0.014$ | $6.51 \pm 0.64$ | $0.801 \pm 0.013$ | $5.76 \pm 0.185$ | $0.817 \pm 0.019$ | $5.29 \pm 0.13$ |
| oocytes-4d | $0.787 \pm 0.020$ | $2.23 \pm 0.09$ | $0.787 \pm 0.038$ | $5.61 \pm 0.26$ | $0.756 \pm 0.021$ | $7.09 \pm 0.27$ | $0.723 \pm 0.006$ | $6.22 \pm 0.12$ | $0.732 \pm 0.030$ | $5.79 \pm 0.04$ |
| oocytes-2f | $0.799 \pm 0.022$ | $1.99 \pm 0.06$ | $0.776 \pm 0.035$ | $5.24 \pm 0.05$ | $0.774 \pm 0.022$ | $6.97 \pm 0.05$ | $0.775 \pm 0.027$ | $5.89 \pm 0.04$ | $0.783 \pm 0.023$ | $5.46 \pm 0.04$ |
| ozone | $0.967 \pm 0.006$ | $3.65 \pm 0.18$ | $0.967 \pm 0.005$ | $6.30 \pm 0.30$ | $0.967 \pm 0.005$ | $6.89 \pm 0.15$ | $0.967 \pm 0.005$ | $7.86 \pm 0.06$ | $0.967 \pm 0.005$ | $6.20 \pm 0.10$ |
| pima | $0.719 \pm 0.019$ | $1.67 \pm 0.11$ | $0.727 \pm 0.018$ | $5.20 \pm 0.29$ | $0.730 \pm 0.031$ | $6.54 \pm 0.21$ | $0.734 \pm 0.025$ | $5.72 \pm 0.07$ | $0.729 \pm 0.015$ | $5.23 \pm 0.02$ |
| spambase | $0.919 \pm 0.007$ | $6.91 \pm 0.34$ | $0.924 \pm 0.004$ | $7.41 \pm 0.04$ | $0.925 \pm 0.005$ | $6.17 \pm 0.07$ | $0.921 \pm 0.003$ | $8.78 \pm 0.20$ | $0.926 \pm 0.005$ | $6.61 \pm 0.11$ |
| statlog-german | $0.761 \pm 0.030$ | $2.22 \pm 0.09$ | $0.755 \pm 0.021$ | $5.84 \pm 0.70$ | $0.756 \pm 0.037$ | $6.39 \pm 0.69$ | $0.753 \pm 0.039$ | $5.89 \pm 0.07$ | $0.758 \pm 0.037$ | $5.48 \pm 0.12$ |
| tic-tac-toe | $0.980 \pm 0.010$ | $1.89 \pm 0.25$ | $0.954 \pm 0.009$ | $4.97 \pm 0.03$ | $0.932 \pm 0.025$ | $6.63 \pm 0.04$ | $0.926 \pm 0.016$ | $5.61 \pm 0.04$ | $0.951 \pm 0.012$ | $5.18 \pm 0.03$ |
| titanic | $0.778 \pm 0.041$ | $0.35 \pm 0.03$ | $0.790 \pm 0.024$ | $5.06 \pm 0.04$ | $0.784 \pm 0.026$ | $6.30 \pm 0.26$ | $0.796 \pm 0.017$ | $6.24 \pm 0.23$ | $0.784 \pm 0.026$ | $5.19 \pm 0.01$ |
| Accuracy/Time | 9/13 | 11/13 | 2/13 | 1/13 | 2/13 | 1/13 | 4/13 | 0/13 | 2/13 | 0/13 |

(SVMs), which find the maximum margin vector. Particularly, we set the zero entries of $\mathbf{d}_i$ as $-1$ and then directly run the SVM to get the maximum margin hidden neurons corresponding to this arrangement. Since the labels are in the form $\{+1, -1\}$ in this case, we do not need additional thresholding as in the previous approach.

**Nonconvex-STE** (Bengio et al., 2013): This is the standard non-convex training algorithm, where the threshold activations is replaced with the identity function during the backward pass.

**STE Variants:** We also benchmark against variants of STE. Specifically, we replace the threshold activation with ReLU (**Nonconvex-ReLU** (Yin et al., 2019a)), Leaky ReLU (**Nonconvex-LReLU** (Xiao et al.)), and clipped ReLU (**Nonconvex-CReLU** (Cai et al., 2017)) during the backward pass.

**Synthetic Datasets:** We compare the performances of **Convex-PI** and **Convex-SVM** trained via (7) with the non-convex heuristic methods mentioned above. We first run each non-convex heuristic for five different initializations and then plot the best performing one in Figure 1. This experiment clearly shows that the non-convex heuristics fail to achieve the globally optimal training performance provided by our convex approaches. For the same setup, we also compare the training and test accuracies for three different regimes, i.e., $n > d$, $n = d$, and $n < d$. As seen in Figure 2, our convex approaches not only globally optimize the training objective but also generalize well on the test data.

**Real Datasets:** In Table 2, we compare the test accuracies of two-layer threshold network trained via our convex formulation in (15), i.e., **Convex-Lasso** and the non-convex heuristics mentioned above. For this experiment, we use CIFAR-10 (Krizhevsky et al., 2014), MNIST (LeCun), and the datasets in the UCI repository (Dua & Graff, 2017) which are preprocessed as in Fernández-Delgado et al. (2014). Here, our convex training approach achieves the highest test accuracy for most of the datasets while the non-convex heuristics perform well only for a few datasets. Therefore, we also validates the good generalization capabilities of the proposed convex training methods on real datasets.

## 6 CONCLUSION

We proved that the training problem of regularized deep threshold networks can be equivalently formulated as a standard convex optimization problem with a fixed data matrix consisting of hyperplane arrangements determined by the data matrix and layer weights. Since the proposed formulation parallels the well studied Lasso model, we have two major advantages over the standard non-convex training methods: **1)** We globally optimize the network without resorting to any optimization heuristic or extensive hyperparameter search (e.g., learning rate schedule and initialization scheme); **2)** We efficiently solve the training problem using specialized solvers for Lasso. We also provided a computational complexity analysis and showed that the proposed convex program can be solved in polynomial-time. Moreover, when a layer width exceeds a certain threshold, a simpler alternative convex formulation can be solved in $\mathcal{O}(n)$. Lastly, as a by product of our analysis, we characterize the recursive process behind the set of hyperplane arrangements for deep networks. Even though this set rapidly grows as the network gets deeper, globally optimizing the resulting Lasso problem still requires polynomial-time complexity for fixed data rank. We also note that the convex analysis proposed in this work is generic in the sense that it can be applied to various architectures including batch normalization (Ergen et al., 2022b), vector output networks (Sahiner et al., 2020; 2021), polynomial activations (Bartan & Pilanci, 2021), GANs (Sahiner et al., 2022a), autoregressive models (Gupta et al., 2021), and Transformers (Ergen et al., 2022a; Sahiner et al., 2022b).

## 7 ACKNOWLEDGEMENTS

This work was partially supported by the National Science Foundation (NSF) under grants ECCS-2037304, DMS-2134248, NSF CAREER award CCF-2236829, the U.S. Army Research Office Early Career Award W911NF-21-1-0242, Stanford Precourt Institute, and the ACCESS – AI Chip Center for Emerging Smart Systems, sponsored by InnoHK funding, Hong Kong SAR.

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

# Appendix

## Table of Contents

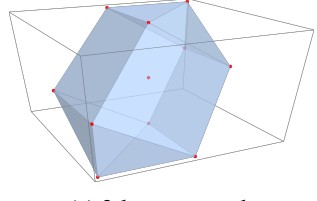

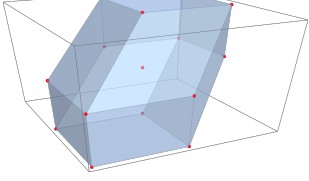

(a) 2-layer network           (b) 3-layer network

Figure 3: The convex hull of $\mathbf{D}^{(1)} = \mathcal{A}(\mathbf{X})$ corresponding to the 2-layer network **(a)** and the convex hull of $\bigsqcup_{|\mathcal{S}|=m_1} \mathcal{A}\left(\mathbf{D}_{\mathcal{S}}^{(1)}\right)$ corresponding to the 3-layer network **(b)** for the data in Example 3.1. Here, we visualize the convex constraint in Proposition 2.4, which is the implicit representation revealed by convex analysis. We observe that the hidden representation space becomes richer with an additional nonlinear layer (from 2-layer to 3-layer), and the resulting symmetry in the convex hull enables the simpler convex formulation (7).

## A   PROOFS OF THE RESULTS IN THE MAIN PAPER

### A.1   LEMMA 2.1

***Proof of Lemma 2.1***. We first restate the original training problem below.

$$\mathcal{P}_2^{\mathrm{noncvx}} = \min_{\mathbf{W}^{(1)}, \mathbf{s}, \mathbf{w}^{(2)}} \frac{1}{2} \| f_{\theta,2}(\mathbf{X}) - \mathbf{y} \|_2^2 + \frac{\beta}{2} \sum_{j=1}^{m} \left( \|\mathbf{w}_j^{(1)}\|_2^2 + s_j^2 + |w_j^{(2)}|^2 \right) . \qquad (19)$$

As already noted in the main paper the loss function is invariant to the norm of $\mathbf{w}_j^{(1)}$ and we may have $\mathbf{w}_j^{(1)} \to 0$ to reduce the regularization cost. Therefore, we can omit the regularization penalty

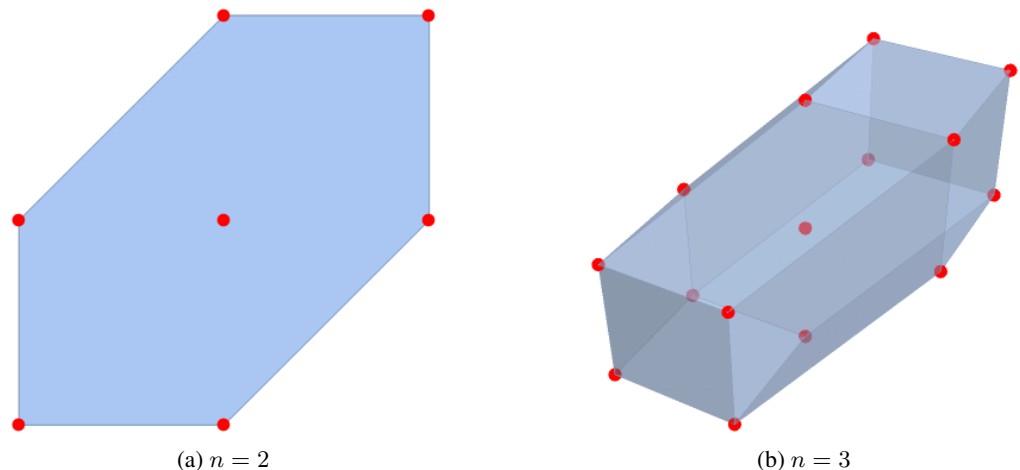

(a) $n = 2$          (b) $n = 3$

Figure 4: Two and three dimensional visualizations of hidden layer representation space of the weight decay regularized threshold networks. Here, we visualize the convex hull in Proposition 2.4, which is the implicit representation space revealed by our analysis.

on $\mathbf{w}_j^{(1)}$ to simplify (19) as

$$\mathcal{P}_2^{\mathrm{noncvx}} = \min_{\mathbf{W}^{(1)}, \mathbf{s}, \mathbf{w}^{(2)}} \frac{1}{2} \|f_{\theta,2}(\mathbf{X}) - \mathbf{y}\|_2^2 + \frac{\beta}{2} \sum_{j=1}^m (s_j^2 + |w_j^{(2)}|^2).$$

We now note that one can rescale the parameters as

$$\bar{s}_j = \alpha_j s_j, \ \bar{w}_j^{(2)} = \frac{w_j^{(2)}}{\alpha_j}$$

without the changing the network's output, i.e.,

$$f_{\bar{\theta},2}(\mathbf{X}) = \sum_{j=1}^m \bar{s}_j \mathbb{1}\{\mathbf{X}\mathbf{w}_j^{(1)}\} \bar{w}_j^{(2)} = \sum_{j=1}^m s_j \mathbb{1}\{\mathbf{X}\mathbf{w}_j^{(1)}\} w_j^{(2)} = f_{\theta,2}(\mathbf{X})$$

where $\alpha_j > 0$. We next note that

$$\frac{\beta}{2} \sum_{j=1}^m (\bar{s}_j^2 + \bar{w}_j^{(2)2}) = \frac{\beta}{2} \sum_{j=1}^m \left( \alpha_j^2 s_j^2 + \frac{|w_j^{(2)}|^2}{\alpha_j^2} \right) \geq \beta \sum_{j=1}^m |s_j||w_j^{(2)}| = \beta \sum_{j=1}^m |\bar{s}_j||\bar{w}_j^{(2)}|,$$

where the equality holds when $\alpha_j = \sqrt{\frac{|w_j^{(2)}|}{|s_j|}}$. Thus, we obtain the following reformulation of the objective function in (19) where the regularization term takes a multiplicative form

$$\mathcal{P}_2^{\mathrm{noncvx}} = \min_{\mathbf{W}^{(1)}, \mathbf{s}, \mathbf{w}^{(2)}} \frac{1}{2} \|f_{\theta,2}(\mathbf{X}) - \mathbf{y}\|_2^2 + \beta \sum_{j=1}^m |s_j||w_j^{(2)}|. \tag{20}$$

Next, we apply a variable change for the new formulation in (20) as follows

$$s_j' := \frac{s_j}{|s_j|}, \quad w_j^{(2)'} := w_j^{(2)}|s_j|.$$

With the variable change above (20) can be equivalently written as

$$\mathcal{P}_2^{\mathrm{noncvx}} = \min_{\substack{\mathbf{W}^{(1)}, \mathbf{w}^{(2)'} \\ \mathbf{s}': |s_j'| = 1}} \frac{1}{2} \|f_{\theta,2}(\mathbf{X}) - \mathbf{y}\|_2^2 + \beta \sum_{j=1}^m |w_j^{(2)'}|. \tag{21}$$

This concludes the proof and yields the following equivalent formulation of (21)

$$\mathcal{P}_2^{\mathrm{noncvx}} = \min_{\substack{\mathbf{W}^{(1)}, \mathbf{w}^{(2)'} \\ \mathbf{s}': |s_j'| = 1}} \frac{1}{2} \|f_{\theta,2}(\mathbf{X}) - \mathbf{y}\|_2^2 + \beta \|\mathbf{w}^{(2)'}\|_1,$$

$\square$

## A.2 PROPOSITION 2.4

***Proof of Proposition 2.4.*** Using the reparameterization for Lasso problems, we rewrite (9) as

$$\min_{w_j^+, w_j^- \geq 0} \sum_{j=1}^{P}(w_j^+ + w_j^-) \quad \text{s.t.} \quad \sum_{j=1}^{P} \mathbf{d}_j(w_j^+ - w_j^-) = \mathbf{y}. \tag{22}$$

We next introduce a slack variable $t \geq 0$ such that (22) can be rewritten as

$$\min_{t \geq 0} \min_{w_j^+, w_j^- \geq 0} t \quad \text{s.t.} \quad \sum_{j=1}^{P} \mathbf{d}_j(w_j^+ - w_j^-) = \mathbf{y}, \sum_{j=1}^{P}(w_j^+ + w_j^-) = t. \tag{23}$$

We now rescale $w_j^+, w_j^-$ with $1/t$ to obtain the following equivalent of (23)

$$\min_{t \geq 0} \min_{w_j^+, w_j^- \geq 0} t \quad \text{s.t.} \quad \sum_{j=1}^{P} \mathbf{d}_j(w_j^+ - w_j^-) = \mathbf{y}/t, \sum_{j=1}^{P}(w_j^+ + w_j^-) = 1. \tag{24}$$

The problem in (24) implies that

$$\exists w_j^+, w_j^- \geq 0 \,\text{s.t.} \sum_{j=1}^{P}(w_j^+ + w_j^-) = 1 \sum_{j=1}^{P} \mathbf{d}_j(w_j^+ - w_j^-) = \mathbf{y}/t \iff \mathbf{y} \in t\text{Conv}\{\pm\mathbf{d}_j, \forall j \in [P]\},$$

where $\text{Conv}$ denotes the convex hull operation. Therefore, (9) can be equivalenly formulated as

$$\min_{t \geq 0} t \quad \text{s.t.} \quad \mathbf{y} \in t\text{Conv}\{\pm\mathbf{d}_j, \forall j \in [P]\}.$$

$\square$

## A.3 THEOREM 2.2

***Proof of Theorem 2.2.*** We first remark that the activations can only be either zero or one, i.e., $\sigma_{s_j}(\mathbf{X}\mathbf{w}_j^{(1)}) \in \{0,1\}^n$, since $|s_j| = 1, \forall j \in [m]$. Therefore, based on Lemma 2.1, we reformulated (4) as

$$\mathcal{P}_2^{\text{noncvx}} = \min_{\substack{\mathbf{d}_{i_j} \in \{\mathbf{d}_1 \dots \mathbf{d}_P\} \\ \mathbf{w}}} \frac{1}{2} \left\| [\mathbf{d}_{i_1}, ..., \mathbf{d}_{i_m}] \mathbf{w} - \mathbf{y} \right\|_2^2 + \beta \|\mathbf{w}\|_1, \tag{25}$$

where we denote the elements of the set $\mathcal{H}(\mathbf{X})$ by $\mathbf{d}_1, \ldots, \mathbf{d}_P \in \{0,1\}^n$ and $P$ is the number of hyperplane arrangements. The above problem is similar to Lasso (Tibshirani, 1996), although it is non-convex due to the discrete variables $\mathbf{d}_{i_1}, ..., \mathbf{d}_{i_m}$. These $m$ hyperplane arrangement patterns are discrete optimization variables along with the coefficient vector $\mathbf{w}$. In fact, the above problem is equivalent to a cardinality-constrained Lasso problem.

We have

$$\mathcal{P}_2^{\text{noncvx}} = \min_{\substack{\mathbf{r} \in \mathbb{R}^n, \mathbf{d}_1, ..., \mathbf{d}_m \in \mathcal{H}(\mathbf{X}), \mathbf{w} \in \mathbb{R}^m \\ \mathbf{r} = [\mathbf{d}_1, ..., \mathbf{d}_m]\mathbf{w} - \mathbf{y}}} \frac{1}{2}\|\mathbf{r}\|_2^2 + \beta\|\mathbf{w}\|_1$$

$$= \min_{\mathbf{r} \in \mathbb{R}^n, \mathbf{d}_1, ..., \mathbf{d}_m \in \mathcal{H}(\mathbf{X}), \mathbf{w} \in \mathbb{R}^m} \max_{\mathbf{z} \in \mathbb{R}^n} \frac{1}{2}\|\mathbf{r}\|_2^2 + \beta\|\mathbf{w}\|_1 + \mathbf{z}^T(\mathbf{r} + \mathbf{y}) - \mathbf{z}^T \sum_{i=1}^{m} \mathbf{d}_i w_i$$

$$\stackrel{(i)}{\geq} \min_{\mathbf{d}_1, ..., \mathbf{d}_m \in \mathcal{H}(\mathbf{X})} \max_{\mathbf{z} \in \mathbb{R}^n} \min_{\mathbf{r} \in \mathbb{R}^n, \mathbf{w} \in \mathbb{R}^m} \mathbf{z}^T\mathbf{y} + \frac{1}{2}\|\mathbf{r}\|_2^2 + \mathbf{z}^T\mathbf{r} + \sum_{i=1}^{m} \beta|w_i| - w_i\mathbf{z}^T\mathbf{d}_i$$

$$= \min_{\mathbf{d}_1, ..., \mathbf{d}_m \in \mathcal{H}(\mathbf{X})} \max_{\substack{\mathbf{z} \in \mathbb{R}^n \\ |\mathbf{z}^T\mathbf{d}_i| \leq \beta, \forall i \in [m]}} \frac{1}{2}\|\mathbf{y}\|_2^2 - \frac{1}{2}\|\mathbf{z} - \mathbf{y}\|_2^2$$

$$\stackrel{(ii)}{\geq} \max_{\substack{\mathbf{z} \in \mathbb{R}^n \\ \max_{\mathbf{d} \in \mathcal{H}(\mathbf{X})} |\mathbf{z}^T\mathbf{d}| \leq \beta}} \frac{1}{2}\|\mathbf{y}\|_2^2 - \frac{1}{2}\|\mathbf{z} - \mathbf{y}\|_2^2$$

$$\stackrel{(iii)}{\geq} \max_{\substack{\mathbf{z} \in \mathbb{R}^n \\ |\mathbf{z}^T\mathbf{d}_i| \leq \beta, \forall i \in [P]}} \frac{1}{2}\|\mathbf{y}\|_2^2 - \frac{1}{2}\|\mathbf{z} - \mathbf{y}\|_2^2 = \mathcal{D}_2^{\text{cvx}},$$

where inequality (i) holds by weak duality, inequality (ii) follows from augmenting the set of constraints $|\mathbf{z}^T\mathbf{d}_i| \leq \beta$ to hold for any sign pattern $\mathbf{d} \in \mathcal{H}(\mathbf{X})$, and inequality (iii) follows from enumerating all possible sign patterns, i.e., $\mathcal{H}(\mathbf{X}) = \{\mathbf{d}_1, \mathbf{d}_2, \ldots, \mathbf{d}_P\}$.

We now prove that strong duality in fact holds, i.e., $\mathcal{P}_2^{\mathrm{noncvx}} = \mathcal{D}_2^{\mathrm{cvx}}$. We first form the Lagrangian of the dual problem $\mathcal{D}_2^{\mathrm{cvx}}$

$$
\begin{aligned}
\mathcal{D}_2^{\mathrm{cvx}} &= \max_{\mathbf{z}\in\mathbb{R}^n} \min_{w_i,w_i'\geq 0} \frac{1}{2}\|\mathbf{y}\|_2^2 - \frac{1}{2}\|\mathbf{z}-\mathbf{y}\|_2^2 + \sum_{i=1}^P \left(w_i(\beta - \mathbf{z}^T\mathbf{d}_i) + w_i'(\beta + \mathbf{z}^T\mathbf{d}_i)\right) \\
&\overset{(i)}{=} \min_{w_i,w_i'\geq 0} \max_{\mathbf{z}\in\mathbb{R}^n} \frac{1}{2}\|\mathbf{y}\|_2^2 - \frac{1}{2}\|\mathbf{z}-\mathbf{y}\|_2^2 + \sum_{i=1}^P \left(w_i(\beta - \mathbf{z}^T\mathbf{d}_i) + w_i'(\beta + \mathbf{z}^T\mathbf{d}_i)\right) \\
&= \min_{w_i,w_i'\geq 0} \frac{1}{2}\left\|\sum_{i=1}^P \mathbf{d}_i(w_i - w_i') - \mathbf{y}\right\|_2^2 + \sum_{i=1}^P \beta\left(w_i + w_i'\right) \\
&= \min_{\mathbf{w}\in\mathbb{R}^P} \frac{1}{2}\|\mathbf{D}\mathbf{w}-\mathbf{y}\|_2^2 + \beta\|\mathbf{w}\|_1 = \mathcal{P}_2^{\mathrm{cvx}}
\end{aligned}
\tag{26}
$$

where, in equality (i) follows from the fact that $\mathcal{D}_2^{\mathrm{cvx}}$ is a convex optimization problem satisfying Slater's conditions so that strong duality holds (Boyd & Vandenberghe, 2004), i.e., $\mathcal{D}_2^{\mathrm{cvx}} = \mathcal{P}_2^{\mathrm{cvx}}$, and the matrix $\mathbf{D} \in \{0,1\}^{n\times P}$ is defined as

$$
\mathbf{D} := [\mathbf{d}_1, \mathbf{d}, \ldots, \mathbf{d}_P].
$$

Now, based on the strong duality results in Ergen & Pilanci (2020); Pilanci & Ergen (2020), there exist a threshold for the number of neurons, i.e., denoted as $m^*$, such that if $m \geq m^*$ then strong duality holds for the original non-convex training problem (25), i.e., $\mathcal{P}_2^{\mathrm{noncvx}} = \mathcal{D}_2^{\mathrm{cvx}} = \mathcal{P}_2^{\mathrm{cvx}}$. Thus, (26) is exactly equivalent to the original non-convex training problem (3).

$\square$

## A.4 THEOREM 2.3

***Proof of Theorem 2.3***. Following the proof of Theorem 2.2, we first note that strong duality holds for the problem in (25), i.e., $\mathcal{P}_2^{\mathrm{noncvx}} = \mathcal{D}_2^{\mathrm{cvx}}$.

Under the assumption that $\mathcal{H}(\mathbf{X}) = \{0,1\}^n$, the dual constraint $\max_{\mathbf{d}\in\mathcal{H}(\mathbf{X})} |\mathbf{z}^T\mathbf{d}| \leq \beta$ is equivalent to $\{\max_{\mathbf{d}\in[0,1]^n} \mathbf{z}^T\mathbf{d} \leq \beta\} \cup \{\max_{\mathbf{d}'\in[0,1]^n} -\mathbf{z}^T\mathbf{d}' \leq \beta\}$. Forming the Lagrangian based on this reformulation of the dual constraint, we have

$$
\begin{aligned}
\mathcal{D}_2^{\mathrm{cvx}} &= \max_{\mathbf{z}\in\mathbb{R}^n} \min_{t,t'\geq 0} \frac{1}{2}\|\mathbf{y}\|_2^2 - \frac{1}{2}\|\mathbf{z}-\mathbf{y}\|_2^2 + t(\beta - \max_{\mathbf{d}\in[0,1]^n}\mathbf{z}^T\mathbf{d}) + t'(\beta + \max_{\mathbf{d}'\in[0,1]^n}\mathbf{z}^T\mathbf{d}') \\
&= \max_{\mathbf{z}\in\mathbb{R}^n} \min_{\substack{t,t'\geq 0 \\ \mathbf{d},\mathbf{d}'\in[0,1]^n}} \frac{1}{2}\|\mathbf{y}\|_2^2 - \frac{1}{2}\|\mathbf{y}-\mathbf{z}\|_2^2 + \mathbf{z}^T(t'\mathbf{d}' - t\mathbf{d}) + \beta(t'+t) \\
&\overset{(i)}{=} \max_{\mathbf{z}\in\mathbb{R}^n} \min_{\substack{t,t'\geq 0 \\ \mathbf{d}\in[0,t]^n,\mathbf{d}'\in[0,t']^n}} \frac{1}{2}\|\mathbf{y}\|_2^2 - \frac{1}{2}\|\mathbf{y}-\mathbf{z}\|_2^2 + \mathbf{z}^T(\mathbf{d}' - \mathbf{d}) + \beta(t'+t) \\
&\overset{(ii)}{=} \min_{\substack{t,t'\geq 0 \\ \mathbf{d}\in[0,t]^n,\mathbf{d}'\in[0,t']^n}} \max_{\mathbf{z}\in\mathbb{R}^n} \frac{1}{2}\|\mathbf{y}\|_2^2 - \frac{1}{2}\|\mathbf{y}-\mathbf{z}\|_2^2 + \mathbf{z}^T(\mathbf{d}' - \mathbf{d}) + \beta(t'+t) \\
&= \min_{\substack{t,t'\geq 0 \\ \mathbf{d}\in[0,t]^n,\mathbf{d}'\in[0,t']^n}} \frac{1}{2}\|\mathbf{d}' - \mathbf{d} - \mathbf{y}\|_2^2 + \beta(t+t')\,.
\end{aligned}
$$

where, in equality (i), we used the change of variables $\mathbf{d} \equiv t\mathbf{d}$ and $\mathbf{d}' \equiv t'\mathbf{d}'$, and, in equality (ii), we used the fact that the objective function $\frac{1}{2}\|\mathbf{y}\|_2^2 - \frac{1}{2}\|\mathbf{y}-\mathbf{z}\|_2^2 + \mathbf{z}^T(\mathbf{d}'-\mathbf{d}) + \beta(t'+t)$ is strongly concave in $\mathbf{z}$ and convex in $(\mathbf{d},\mathbf{d}',t,t')$ and the constraints are linear, so that strong duality holds and we can switch the order of minimization and maximization.

Given a feasible point $(\mathbf{d}, \mathbf{d}', t, t')$, i.e., $\mathbf{d} \in [0, t]^n$ and $\mathbf{d}' \in [0, t']^n$, we set $\boldsymbol{\delta} = \mathbf{d}' - \mathbf{d}$. Note that $\|(\boldsymbol{\delta})_+\|_\infty \leq \|(\mathbf{d}')_+\|_\infty = \|\mathbf{d}'\|_\infty \leq t'$. Similarly, $\|(-\boldsymbol{\delta})_+\|_\infty \leq t$. This implies that

$$\min_{\substack{t, t' \geq 0 \\ \mathbf{d} \in [0,t]^n, \mathbf{d}' \in [0,t']^n}} \frac{1}{2}\|\mathbf{d}' - \mathbf{d} - \mathbf{y}\|_2^2 + \beta(t + t') \geq \min_{\boldsymbol{\delta} \in \mathbb{R}^n} \frac{1}{2}\|\boldsymbol{\delta} - \mathbf{y}\|_2^2 + \beta(\|(\boldsymbol{\delta})_+\|_\infty + \|(-\boldsymbol{\delta})_+\|_\infty) = \mathcal{P}_{v2}^{\mathrm{cvx}}.$$

Conversely, given $\boldsymbol{\delta} \in \mathbb{R}^n$, we set $\mathbf{d} = (\boldsymbol{\delta})_+$, $t = \|\mathbf{d}\|_\infty$, $\mathbf{d}' = (-\boldsymbol{\delta})_+$ and $t' = \|\mathbf{d}'\|_\infty$. It holds that $(\mathbf{d}, \mathbf{d}', t, t')$ is feasible with same objective value, and consequently, the above inequality is an equality, i.e., $\mathcal{D}_2^{\mathrm{cvx}} = \mathcal{P}_{v2}^{\mathrm{cvx}}$.

**Optimal threshold network construction:** We now show how to construct an optimal threshold network given an optimal solution to the convex problem (7). Let $\boldsymbol{\delta} \in \mathbb{R}^n$ be an optimal solution. Set $\mathbf{d} = (\boldsymbol{\delta})_+$ and $\mathbf{d}' = (-\boldsymbol{\delta})_+$. We have $\mathbf{d} \in [0, \|\mathbf{d}\|_\infty]^n$ and $\mathbf{d}' \in [0, \|\mathbf{d}'\|_\infty]^n$. It is easy to show that we can transform $\boldsymbol{\delta}$ such that, for each index $i \in [n]$, either the $i$-th coordinate of $\mathbf{d}$ is active or the $i$-th coordinate of $\mathbf{d}'$ is active. Therefore, by Caratheodory's theorem, there exist $n_+, n_- \geq 1$ such that $n_- + n_+ \leq n$, and $\mathbf{d}_1, \ldots, \mathbf{d}_{n_++1} \in \{0, 1\}^n$ and $\gamma_1, \ldots, \gamma_{n_++1} \geq 0$ such that $\sum_{i=1}^{n_++1} \gamma_i = 1$ and $\mathbf{d} = \|\mathbf{d}\|_\infty \sum_{i=1}^{n_++1} \gamma_i \mathbf{d}_i$, and, $\mathbf{d}'_1, \ldots, \mathbf{d}'_{n_-+1} \in \{0, 1\}^n$ and $\gamma'_1, \ldots, \gamma'_{n_-+1} \geq 0$ such that $\sum_{i=1}^{n_-+1} \gamma'_i = 1$ and $\mathbf{d}' = \|\mathbf{d}'\|_\infty \sum_{i=1}^{n_-+1} \gamma'_i \mathbf{d}'_i$, with $n_- + n_+ \leq n$. Then, we can pick $\mathbf{w}_1^{(1)}, \ldots, \mathbf{w}_{n_++1}^{(1)}$, $w_1^{(2)}, \ldots, w_{n_++1}^{(2)}, s_1, \ldots, s_{n_++1}$ such that $\mathbb{1}\{\mathbf{X}\mathbf{w}_i^{(1)} \geq 0\} = \mathbf{d}_i$, $w_i^{(1)} = \|\mathbf{d}\|_\infty \gamma_i$ and $s_i = -1$, and, $\mathbf{w}_{n_++2}^{(1)}, \ldots, \mathbf{w}_{n_++n_-+2}^{(1)}$, $w_{n_++2}^{(2)}, \ldots, w_{n_++n_-+2}^{(2)}, s_{n_++2}, \ldots, s_{n_++n_-+2}$ such that $\mathbb{1}\{\mathbf{X}\mathbf{w}_i^{(1)} \geq 0\} = \mathbf{d}'_{i-(n_++1)}$, $\alpha_i = \|\mathbf{d}'\|_\infty \gamma_{i-(n_++1)}$ and $s_i = 1$. Note that, given $\boldsymbol{\delta} \in \mathbb{R}^n$, finding the corresponding $\mathbf{d}_1, \ldots, \mathbf{d}_{n_++1}, \gamma_1, \ldots, \gamma_{n_++1}$ and $\mathbf{d}'_1, \ldots, \mathbf{d}'_{n_++1}, \gamma'_1, \ldots, \gamma'_{n_++1}$ takes time $\mathcal{O}(n_+ + n_+) = \mathcal{O}(n)$. □

## A.5 LEMMA 3.1

***Proof of Lemma 3.1.*** We first restate the standard weight decay regularized training problem below

$$\min_{\theta \in \Theta} \frac{1}{2}\|f_{\theta,L}(\mathbf{X}) - \mathbf{y}\|_2^2 + \frac{\beta}{2} \sum_{k=1}^{m_{L-1}} \sum_{l=1}^{L} (\|\mathbf{W}_k^{(l)}\|_F^2 + \|\mathbf{s}_k^{(l)}\|_2^2). \tag{27}$$

Then, we remark that the loss function in (27), i.e., $\frac{1}{2}\|f_{\theta,L}(\mathbf{X}) - \mathbf{y}\|_2^2$ is invariant to the norms of hidden layer weights $\{\{\mathbf{W}_k^{(l)}\}_{l=1}^{L-1}\}_{k=1}^{m_{L-1}}$ and amplitude parameters $\{\{\mathbf{s}_k^{(l)}\}_{l=1}^{L-2}\}_{k=1}^{m_{L-1}}$. Therefore, (27) can be rewritten as

$$\min_{\theta \in \Theta} \frac{1}{2}\|f_{\theta,L}(\mathbf{X}) - \mathbf{y}\|_2^2 + \frac{\beta}{2} \sum_{k=1}^{m_{L-1}} \left((w_k^{(L)})^2 + (s_k^{(L-1)})^2\right). \tag{28}$$

Then, we can directly apply the scaling in the proof of Lemma 2.1 to obtain the claimed $\ell_1$-norm regularized optimization problem. □

## A.6 THEOREM 3.2

***Proof of Theorem 3.2.*** We first remark that the last hidden layer activations can only be either zero or one, i.e., $\mathbf{X}_k^{(L-1)} \in \{0, 1\}^{n \times m_{L-1}}$, since $|s_k^{(L-1)}| = 1, \forall k \in [m_{L-1}]$. Therefore, based on Lemma 3.1, we reformulated (12) as

$$\mathcal{P}_L^{\mathrm{noncvx}} = \min_{\substack{\mathbf{d}_{i_j} \in \mathcal{H}_L(\mathbf{X}) \\ \mathbf{w}}} \frac{1}{2}\left\|\left[\mathbf{d}_{i_1}, \ldots, \mathbf{d}_{i_{m_{L-1}}}\right]\mathbf{w} - \mathbf{y}\right\|_2^2 + \beta\|\mathbf{w}\|_1, \tag{29}$$

As in (25), the above problem is similar to Lasso, although it is non-convex due to the discrete variables $\mathbf{d}_{i_1}, \ldots, \mathbf{d}_{i_{m_{L-1}}}$. These $m_{L-1}$ hyperplane arrangement patterns are discrete optimization variables along with the coefficient vector $\mathbf{w}$.

We now note that (29) has following dual problem

$$\mathcal{D}_L^{\mathrm{cvx}} = \mathcal{D}_2^{\mathrm{cvx}} = \max_{\max_{\mathbf{d} \in \mathcal{H}_L(\mathbf{X})} |\mathbf{z}^T \mathbf{d}| \leq \beta} -\frac{1}{2}\|\mathbf{z} - \mathbf{y}\|_2^2 + \frac{1}{2}\|\mathbf{y}\|_2^2. \tag{30}$$

Since the above dual form is the same with the dual problem of two-layer networks, i.e., $\mathcal{D}_L^{\text{cvx}} = \mathcal{D}_2^{\text{cvx}}$, we directly follow the proof of Theorem 2.2 to obtain the following bidual formulation.

$$\mathcal{D}_L^{\text{cvx}} = \mathcal{P}_L^{\text{cvx}} = \min_{\mathbf{w} \in \mathbb{R}^{P_{L-1}}} \frac{1}{2} \left\| \mathbf{D}^{(L-1)}\mathbf{w} - \mathbf{y} \right\|_2^2 + \beta \|\mathbf{w}\|_1,$$

where $\mathbf{D}^{(L-1)} = [\mathbf{d}_1, \mathbf{d}, \dots, \mathbf{d}_{P_{L-1}}]$.

Hence, based on the strong duality results in Ergen & Pilanci (2020); Pilanci & Ergen (2020); Ergen & Pilanci (2021c), there exist a threshold for the number of neurons, i.e., denoted as $m^*$, such that if $m_{L-1} \geq m^*$ then strong duality holds for the original non-convex training problem (25), i.e., $\mathcal{P}_L^{\text{noncvx}} = \mathcal{D}_L^{\text{cvx}} = \mathcal{P}_L^{\text{cvx}}$. $\qquad\qquad\qquad\qquad\qquad\qquad\qquad\qquad\qquad\qquad\qquad\qquad\qquad\qquad\square$

## A.7  LEMMA 3.3

***Proof of Lemma 3.3***. Based on the construction in (Bartlett et al., 2019), a deep network with $L$ layers and $W$ parameters, where $W = \sum_{i=1}^{L} m_{i-1}m_i$ in our notation, can shatter a dataset of the size $C_1 W L \log(W)$, where $C_1$ is a constant. Based on this complexity results, if we choose $m_l = m = C\sqrt{n}/L, \forall l \in [L]$, the number of samples that can be shattered by an $L$-layer threshold network is

$$\text{\# of data samples that can be shattered} = C_1 W L \log(W)$$
$$= C_1 m^2 L^2 \log(m^2 L)$$
$$= C_1 C^2 n \log\left(\frac{C^2 n}{L}\right)$$
$$= n \log\left(\frac{n}{L C_1}\right) \gg n$$

provided that $n \gg L$, which is usually holds in practice and we choose the constant $C$ such that $C = 1/C_1^2$. Therefore, as we make the architecture deeper, we can in fact improve the assumption $m \geq n + 2$ assumption in the two-layer networks case by benefiting from the depth.

$\qquad\qquad\qquad\qquad\qquad\qquad\qquad\qquad\qquad\qquad\qquad\qquad\qquad\qquad\qquad\qquad\qquad\qquad\qquad\qquad\square$

## A.8  LEMMA 3.5

***Proof of Proposition 3.5***. Let us start with the $L = 2$ case for which hyperplane arrangement set can described as detailed in (5)

$$\mathcal{H}_2(\mathbf{X}) = \{\mathbb{1}\{\mathbf{X}\mathbf{w}^{(1)} \geq 0\} : \theta \in \Theta\}$$
$$= \mathcal{H}(\mathbf{X})$$
$$= \{\mathbf{d}_1, \dots, \mathbf{d}_{P_1}\} \subset \{0,1\}^n.$$

Now, in order to construct $\mathcal{H}_3(\mathbf{X})$ (with $m_{L-1} = 1$ so we drop the index $k$), we need to choose $m_1$ arrangements from $\mathcal{H}_2(\mathbf{X})$ and then consider all hyperplane arrangements for each of these choices. Particularly, since $|\mathcal{H}_2(\mathbf{X})| = P_1$, where

$$P_1 \leq 2 \sum_{k=0}^{r-1} \binom{n-1}{k} \leq 2r \left(\frac{e(n-1)}{r}\right)^r \approx \mathcal{O}(n^r)$$

from (8), we have $\binom{P_1}{m_1}$ choices and each of them yields a different activation matrix denoted as $\mathbf{X}_i^{(1)} \in \mathbb{R}^{n \times m_1}$. Based on the upperbound (8), each $\mathbf{X}_i^{(1)}$ can generate $\mathcal{O}(n^{m_1})$ patterns.

Therefore, overall, the set of possible arrangement in the second layer is as follows

$$\mathcal{H}_3(\mathbf{X}) = \bigcup_{i=1}^{\binom{P_1}{m_1}} \{\mathbb{1}\{\mathbf{X}^{(1)}\mathbf{w}^{(2)} \geq 0\} : \theta \in \Theta\}\Big|_{\mathbf{X}^{(1)} = \mathbf{X}_i^{(1)}}$$

which implies that

$$|\mathcal{H}_3(\mathbf{X})| = P_2 \lesssim \binom{P_1}{m_1} \mathcal{O}(n^{m_1}) \approx \mathcal{O}(n^{m_1 r}).$$

This analysis can be recursively extended to the $l^{th}$ layer where the hyperplane arrangement set and the corresponding cardinality values can be computed as follows

$$\mathcal{H}_l(\mathbf{X}) = \bigcup_{i=1}^{\binom{P_{l-2}}{m_{l-2}}} \{\mathbb{1}\{\mathbf{X}^{(l-2)}\mathbf{w}^{(l-1)} \geq 0\} : \theta \in \Theta\}\Big|_{\mathbf{X}^{(l-2)}=\mathbf{X}_i^{(l-2)}}$$

$$|\mathcal{H}_l(\mathbf{X})| = P_{l-1} \lesssim \binom{P_{l-2}}{m_{l-2}} \mathcal{O}(n^{m_{l-2}}) \approx \mathcal{O}(n^r \prod_{k=1}^{l-2} m_k).$$

$\square$

## A.9 COROLLARY 3.4

***Proof of Corollary 3.4.*** We first remark that the last hidden layer activations can only be either zero or one, i.e., $\mathbf{X}_k^{(L-1)} \in \{0,1\}^{n \times m_{L-1}}$, since $|s_k^{(L-1)}| = 1, \forall k \in [m_{L-1}]$. Therefore, based on Lemma 3.1, we reformulated (12) as

$$\min_{\substack{\mathbf{d}_{i_j} \in \mathcal{H}_L(\mathbf{X}) \\ \mathbf{w}}} \frac{1}{2} \left\| \left[\mathbf{d}_{i_1}, ..., \mathbf{d}_{i_{m_{L-1}}}\right] \mathbf{w} - \mathbf{y} \right\|_2^2 + \beta \|\mathbf{w}\|_1 , \tag{31}$$

As in (25), the above problem is similar to Lasso, although it is non-convex due to the discrete variables $\mathbf{d}_{i_1}, ..., \mathbf{d}_{i_{m_{L-1}}}$. These $m_{L-1}$ hyperplane arrangement patterns are discrete optimization variables along with the coefficient vector $\mathbf{w}$.

We now note that (31) has following dual problem

$$\max_{\max_{\mathbf{d} \in \mathcal{H}_L(\mathbf{X})} |\mathbf{z}^T \mathbf{d}| \leq \beta} -\frac{1}{2}\|\mathbf{z} - \mathbf{y}\|_2^2 + \frac{1}{2}\|\mathbf{y}\|_2^2.$$

Since $\mathcal{H}_L(\mathbf{X}) = \{0,1\}^n$ by Lemma 3.3, all the steps in the proof of Theorem 2.3 directly follow.

**Optimal deep threshold network construction:** For the last two layers' weights, we exactly follow the weight construction procedure in the Proof of Theorem 2.3 as detailed below.

Let $\boldsymbol{\delta} \in \mathbb{R}^n$ be an optimal solution. Set $\mathbf{d} = (\boldsymbol{\delta})_+$ and $\mathbf{d}' = (-\boldsymbol{\delta})_+$. We have $\mathbf{d} \in [0, \|\mathbf{d}\|_\infty]^n$ and $\mathbf{d}' \in [0, \|\mathbf{d}'\|_\infty]^n$. It is easy to show that we can transform $\boldsymbol{\delta}$ such that, for each index $i \in [n]$, either the $i$-th coordinate of $\mathbf{d}$ is active or the $i$-th coordinate of $\mathbf{d}'$ is active. Therefore, by Caratheodory's theorem, there exist $n_+, n_- \geq 1$ such that $n_- + n_+ \leq n$, and $\mathbf{d}_1, \ldots, \mathbf{d}_{n_++1} \in \{0,1\}^n$ and $\gamma_1, \ldots, \gamma_{n_++1} \geq 0$ such that $\sum_{i=1}^{n_++1} \gamma_i = 1$ and $\mathbf{d} = \|\mathbf{d}\|_\infty \sum_{i=1}^{n_++1} \gamma_i \mathbf{d}_i$, and, $\mathbf{d}'_1, \ldots, \mathbf{d}'_{n_-+1} \in \{0,1\}^n$ and $\gamma'_1, \ldots, \gamma'_{n_-+1} \geq 0$ such that $\sum_{i=1}^{n_-+1} \gamma'_i = 1$ and $\mathbf{d}' = \|\mathbf{d}'\|_\infty \sum_{i=1}^{n_-+1} \gamma'_i \mathbf{d}'_i$, with $n_- + n_+ \leq n$. Then, we can pick $\mathbf{w}_1^{(L-1)}, \ldots, \mathbf{w}_{n_++1}^{(L-1)}, w_1^{(L)}, \ldots, w_{n_++1}^{(L)}, s_1^{(L-1)}, \ldots, s_{n_++1}^{(L-1)}$ such that $\mathbb{1}\{\mathbf{X}^{(L-2)}\mathbf{w}_i^{(L-1)} \geq 0\} = \mathbf{d}_i, \mathbf{w}_i^{(L-1)} = \|\mathbf{d}\|_\infty \gamma_i$ and $s_i = -1$, and, $\mathbf{w}_{n_++2}^{(L-1)}, \ldots, \mathbf{w}_{n_++n_-+2}^{(1)}, w_{n_++2}^{(L)}, \ldots, w_{n_++n_-+2}^{(L)}, s_{n_++2}^{(L-1)}, \ldots, s_{n_++n_-+2}^{(L-1)}$ such that $\mathbb{1}\{\mathbf{X}^{(L-2)}\mathbf{w}_i^{(L-1)} \geq 0\} = \mathbf{d}'_{i-(n_++1)}$, $\alpha_i = \|\mathbf{d}'\|_\infty \gamma_{i-(n_++1)}$ and $s_i = 1$. Note that, given $\boldsymbol{\delta} \in \mathbb{R}^n$, finding the corresponding $\mathbf{d}_1, \ldots, \mathbf{d}_{n_++1}, \gamma_1, \ldots, \gamma_{n_++1}$ and $\mathbf{d}'_1, \ldots, \mathbf{d}'_{n_-+1}, \gamma'_1, \ldots, \gamma'_{n_++1}$ takes time $\mathcal{O}(n_- + n_+) = \mathcal{O}(n)$.

Then, the rest of the layer weights can be reconstructed using the construction procedure detailed in (Bartlett et al., 2019).

$\square$

### A.10 COROLLARY 4.1

***Proof of Corollary 4.1.*** We first apply the scaling in Lemma 3.1 and then follow the same steps in the proof of Theorem 2.3 to get the following dual problem

$$\max_{\max_{\mathbf{d}\in\mathcal{H}(\mathbf{X})}|\mathbf{z}^T\mathbf{d}|\leq\beta} -\mathcal{L}^*(\mathbf{z}) \,, \tag{32}$$

where $\mathcal{L}^*$ is the Fenchel conjugate of $\mathcal{L}$ and defined as

$$\mathcal{L}^*(\mathbf{z}) := \max_{\mathbf{v}} \mathbf{v}^T\mathbf{z} - \mathcal{L}(\mathbf{v},\mathbf{y}).$$

Therefore, we obtain a generic version of the dual problem in (30), where we can arbitrarily choose the network's depth and the convex loss function. Then the rest of the proof directly follows from Theorem 3.2 and Corollary 3.4. □

## B ADDITIONAL EXPERIMENTS AND DETAILS

In this section, we present additional numerical experiments and further experimental details that are not presented in the main paper due to the page limit.

We first note that all of the experiments in the paper are run on a single laptop with Intel(R) Core(TM) i7-7700HQ CPU and 16GB of RAM.

### B.1 EXPERIMENT IN FIGURE 5

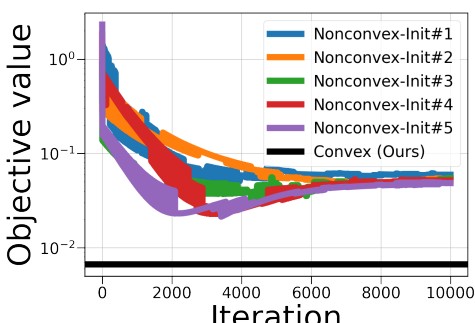

Figure 5: Comparison of our convex training method in (6) with standard non-convex training heuristic for threshold networks, known as Straight-Through Estimator (STE) (Bengio et al., 2013). For the non-convex heuristic, we repeat the training process using 5 independent initializations, however, all trials fail to converge to the global minimum obtained by our convex optimal method, and lack stability. We provide experimental details in Appendix B.

For the experiment in Figure 5, we consider a simple one dimensional experiment, where the data matrix is $\mathbf{X} = [-2,-1,0,1,2]^T$. Using this data matrix, we generate the corresponding labels $\mathbf{y}$ by simply forward propagating the data through randomly initialized two-layer threshold networks with $m_1 = 2$ neurons as described in (2). We then run our convex training method in (6) and the non-convex training heuristic STE (Bengio et al., 2013) on the objective with the regularization coefficient $\beta = 1e-2$. For a fair comparison, we used $P_1 = 24$ for the convex method and $m_1 = 24$ for STE. We also tune the learning rate of STE by performing a grid search on the set $\{5e-1, 1e-1, 5e-2, 1e-2, 5e-3, 1e-3\}$. As illustrated in Figure 5, the non-convex training heuristic STE fails to achieve the global minimum obtained by our convex training algorithm for 5 different initialization trials.

### B.2 EXPERIMENT IN TABLE 2

Here, we provide a test performance comparison on on CIFAR-10 (Krizhevsky et al., 2014), MNIST (LeCun), and the datasets taken from UCI Machine Learning Repository (Dua & Graff, 2017), where

Table 3: Dataset sizes for the experiments in Table 2.

| Dataset | $n$ | $d$ |
|---|---|---|
| CIFAR-10 | 10000 | 3072 |
| MNIST | 12665 | 784 |
| bank | 4521 | 16 |
| chess-krvkp | 3196 | 36 |
| mammographic | 961 | 5 |
| oocytes-4d | 1022 | 41 |
| oocytes-2f | 912 | 25 |
| ozone | 2536 | 72 |
| pima | 768 | 8 |
| spambase | 4601 | 57 |
| statlog-german | 1000 | 24 |
| tic-tac-toe | 958 | 9 |
| titanic | 2201 | 3 |

we follow the preprocesing in Fernández-Delgado et al. (2014) such that $750 \leq n \leq 5000$ (see Table 3 for the exact dataset sizes). We particularly consider a conventional binary classification framework with two-layer networks and performed simulations over multiple seeds and compare the mean/std of the test accuracies of non-convex heuristics trained with SGDnamely **Nonconvex-STE** (Bengio et al., 2013), **Nonconvex-ReLU** (Yin et al., 2019a), **Nonconvex-LReLU** (Xiao et al.) and **Nonconvex-CReLU** (Cai et al., 2017), with our convex program in (6), i.e., **Convex-Lasso**. For the non-convex training, we use the SGD optimizer. We also use the $80\% - 20\%$ splitting ratio for the training and test sets of the UCI datasets. We tune the regularization coefficient $\beta$ and the learning rate $\mu$ for the non-convex program by performing a grid search on the sets $\beta_{\text{list}} = \{1e-6, 1e-3, 1e-2, 1e-1, 0.5, 1, 5\}$ and $\mu_{\text{list}} = \{1e-3, 5e-3, 1e-2, 1e-1\}$, respectively. In all experiments, we also decayed the selected learning rate systematically using PyTorch's (Paszke et al., 2019) scheduler ReduceLROnPlateau. Moreover, we choose the number of neurons, number of epochs (ne), batch size (bs) as $m = 1000$, bs $= 5000$, bs $= n$, respectively. Our convex approach achieves highest test accuracy for precisely 9 of 13 datasets whereas the best non-convex heuristic achieves the highest test accuracy only for 4 datasets. This experiment verifies that our convex training approach not only globally optimize the training objective but also usually generalizes well on the test data. In addition, our convex training approach is shown to be significantly more time efficient than standard non-convex training.

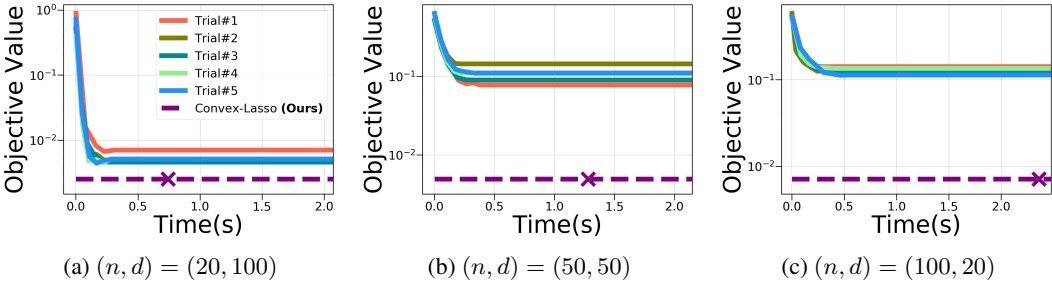

(a) $(n, d) = (20, 100)$          (b) $(n, d) = (50, 50)$          (c) $(n, d) = (100, 20)$

Figure 6: Training performance comparison of our convex program for two-layer networks in (6) with four standard non-convex training heuristics (STE and its variants in Section 5). To generate the dataset, we randomly initialize a two-layer network and then sample a random i.i.d. Gaussian data matrix $\mathbf{X} \in \mathbb{R}^{n \times d}$. Then, we set the labels as $\mathbf{y} = \text{sgn}(\tanh(\mathbf{X}\mathbf{W}^{(1)})\mathbf{w}^{(2)})$ where sgn and tanh are the sign and hyperbolic tangent functions. We specifically choose the regularization coefficient $\beta = 1e-3$ and run the training algorithms with $m = 1000$ neurons for different $(n, d)$ combinations. For each non-convex method, we repeat the training with 5 different initialization and then plot the best performing non-convex method for each initialization trial. In each case, our convex training algorithm achieves significantly lower objective than all the non-convex heuristics. We also indicate the time taken to solve the convex program with a marker.

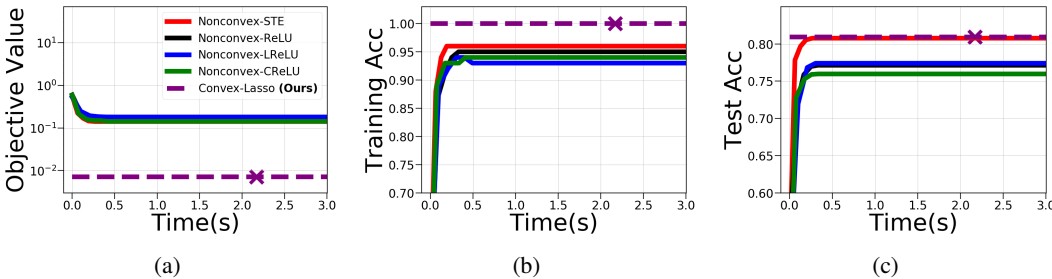

Figure 7: In this figure, we compare the classification performance of the simulation in Figure 6c for a single initialization trial, where $n = 100$ and $d = 20$. This experiment shows that our convex training approach not only provides the optimal training performance but also generalizes on the test data as demonstrated in **(c)**.

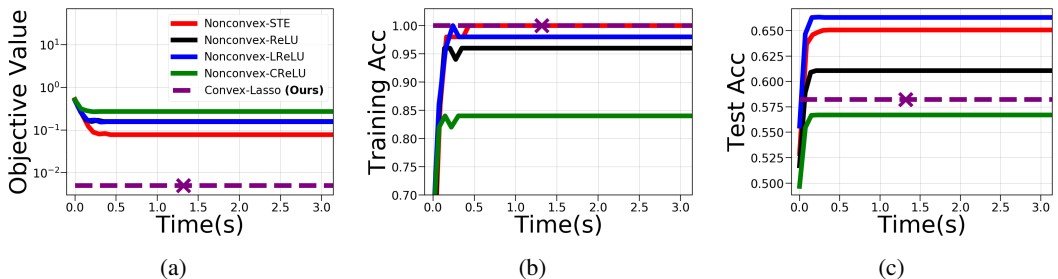

Figure 8: The setup for these figures is completely the same with Figure 7 except that we consider the $(n, d) = (50, 50)$ case in Figure 6b. Unlike Figure 7, here even though our convex training approach provides the optimal training, the non-convex heuristic methods that are stuck at a local minimum yield a better test accuracy. This is due to the fact that we have less data samples with higher dimensionality compared to Figure 7.

### B.3 TWO-LAYER EXPERIMENTS IN FIGURE 6, 7, 8, AND 9

Here, we compare two-layer threshold network training performance of our convex program (6), which we call **Convex-Lasso**, with standard non-convex training heuristics, namely **Nonconvex-STE** (Bengio et al., 2013), **Nonconvex-ReLU** (Yin et al., 2019a), **Nonconvex-LReLU** (Xiao et al.) and **Nonconvex-CReLU** (Cai et al., 2017). For the non-convex heuristics, we train a standard two-layer threshold network with the SGD optimizer. In all experiments, learning rates are initialized to be $0.01$ and they are decayed systematically using PyTorch's (Paszke et al., 2019) scheduler `ReduceLROnPlateau`. To generate a dataset, we first sample an i.i.d. Gaussian data matrix $\mathbf{X}$ and then obtain the corresponding labels as $\mathbf{y} = \text{sgn}(\tanh(\mathbf{X}\mathbf{W}^{(1)})\mathbf{w}^{(2)})$ where sgn and tanh are the sign and hyperbolic tangent functions, respectively. Here, we denote the the ground truth parameters as $\mathbf{W}^{(1)} \in \mathbb{R}^{d \times m^*}$ and $\mathbf{w}^{(2)} \in \mathbb{R}^{m^*}$, where $m^* = 20$ is the number of neurons in the ground truth model. Notice that we use tanh and sign in the ground truth model to have balanced label distribution $\mathbf{y} \in \{+1, -1\}^n$. We also note that for all the experiments, we choose the regularization coefficient as $\beta = 1e - 3$.

We now emphasize that to have a fair comparison with the non-convex heuristics, we first randomly sample a small subset of hyperplane arrangements and then solve (6) with this fixed small subset. Specifically, instead of enumerating all possible arrangements $\{\mathbf{d}_i\}_{i=1}^{P}$, we randomly sample a subset $\{\mathbf{d}_{i_j}\}_{j=1}^{m}$ to have a fair comparison with the non-convex neural network training with $m$ hidden neurons. So, **Convex-Lasso** is an approximate way to solve the convex program (6) yet it still performs extremely well in our experiments. We also note that the other convex approaches **Convex-PI** and **Convex-SVM** exactly solve the proposed convex programs.

In Figure 6, we compare training performances. We particularly solve the convex optimization problem (6) once. However, for the non-convex training heuristics, we try five different initializations.

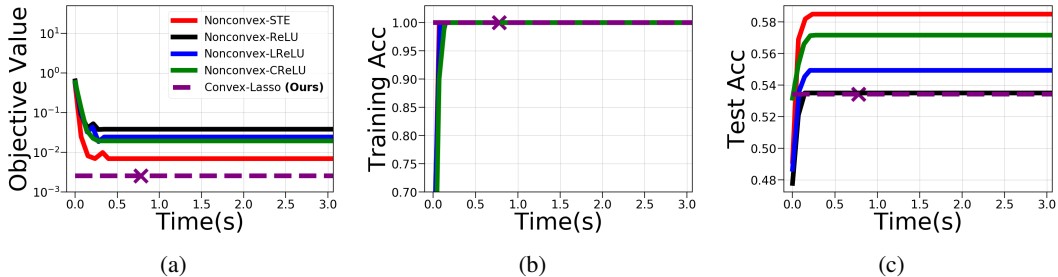

(a)                                      (b)                                      (c)

Figure 9: The setup for these figures is completely the same with Figure 7 except that we consider the $(n, d) = (20, 100)$ case in Figure 6a. As in Figure 8, even though the non-convex heuristic training methods fail to globally optimize the objective, they yield higher test accuracies than the convex program due to the low data regime ($n \leq d$).

We then select the trial with the lowest objective value to plot. In the figure, we plot the objective value defined in (3). As it can be seen from the figures, **Convex-Lasso** achieves much lower objective value than the non-convex heuristics in three different regimes, i.e., overparameterized ($n < d$), underparameterized ($n > d$) and moderately parameterized (($n = d$)). This is mainly because of the non-convex and heuristic nature of the standard optimizers.

Moreover, we illustrate training and test accuracies of these training algorithms in Figure 7, 8, and 9 for different $(n, d)$ combinations. To generate the test data with $3000$, samples we use the same ground truth model defined above. Here we observe that in some cases even though **Convex-Lasso** provides a better training performance, it might yield worse test accuracies than the non-convex heuristic especially in the low data regime ($n \leq d$). Therefore, understanding the generalization performance and optimal weight reconstruction (discussed in Section 2.4) remains to be an open problem.

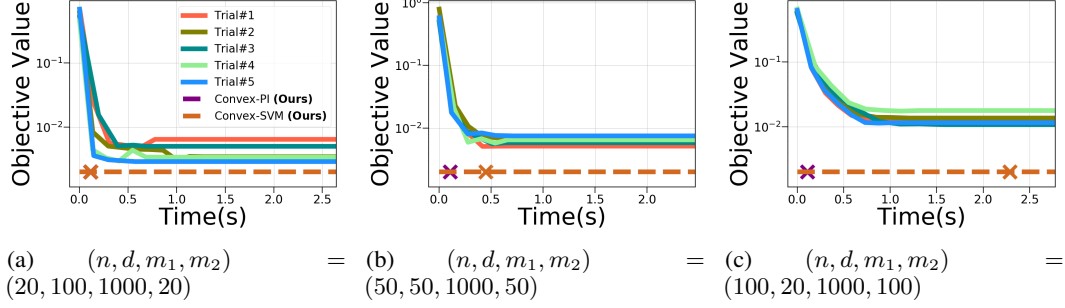

(a)      $(n, d, m_1, m_2)$     = (b)     $(n, d, m_1, m_2)$     = (c)     $(n, d, m_1, m_2)$     =
$(20, 100, 1000, 20)$                $(50, 50, 1000, 50)$             $(100, 20, 1000, 100)$

Figure 10: Training comparison of our convex program in (7) with three-layer threshold networks trained with the non-convex heuristics. Here, we directly follow the setup in Figure 6 except the following differences. This time we randomly initialize a three-layer network and then sample i.i.d. Gaussian data matrix. We then set the labels as $\mathbf{y} = \text{sgn}(\tanh(\tanh(\mathbf{XW}^{(1)})\mathbf{W}^{(2)})\mathbf{w}^{(3)})$. To have our convex formulation, we require $\mathcal{H}(\mathbf{X})$ to be complete. Thus, we first use a random representation matrix $\mathbf{H} \in \mathbb{R}^{d \times M}$, where $M = 1000$, and then apply the transformation $\tilde{\mathbf{X}} = \sigma(\mathbf{XH})$. We also apply this transformation to the non-convex methods as if we train a two-layer networks on the modified data matrix $\tilde{\mathbf{X}}$. We also indicate the time taken to solve the convex programs with markers. We again observe that our convex training approach achieves lower objective value than all the non-convex heuristic training methods in all initialization trials.

## B.4   Three-layer experiments with a representation matrix in Figure 10, 11, 12, and 13

We also compare our alternative convex formulation in (7) with non-convex approaches for three-layer network training. To do so we first generate a dataset via the following ground truth model $\mathbf{y} =$

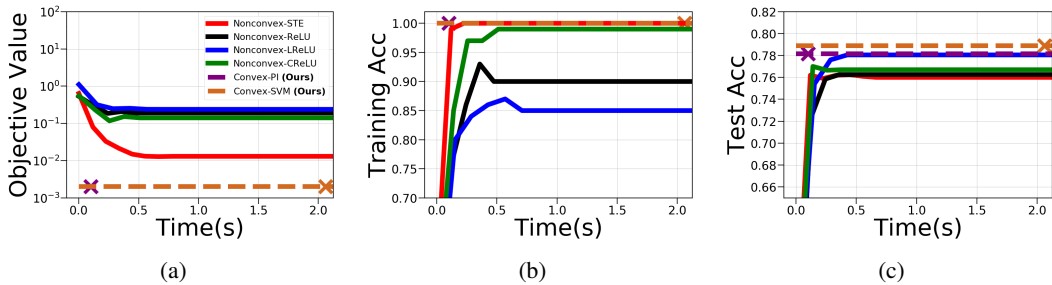

Figure 11: In this figure, we compare the classification performance of the simulation in Figure 10c for a single initialization trial, where $(n, d) = (100, 20)$. This experiment shows that our convex training approach not only provides the optimal training performance but also generalizes on the test data as demonstrated in **(c)**.

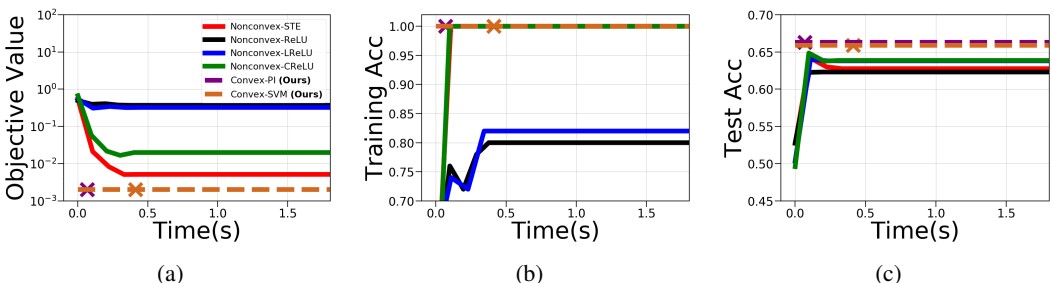

Figure 12: The setup for these figures is completely the same with Figure 11 except that we consider the $(n, d) = (50, 50)$ case in Figure 10b. This experiment shows that when we have all possible hyperplane arrangements, our convex training approach generalizes better even in the low data regime $(n \leq d)$.

$\text{sgn}(\tanh(\tanh(\mathbf{X}\mathbf{W}^{(1)})\mathbf{W}^{(2)})\mathbf{w}^{(3)})$, where $\mathbf{W}^{(1)} \in \mathbb{R}^{d \times m_1^*}$, $\mathbf{W}^{(2)} \in \mathbb{R}^{m_1^* \times m_2^*}$ and $\mathbf{w}^{(3)} \in \mathbb{R}^{m_2^*}$ and we choose $m_1^* = m_2^* = 20$ for all the experiments. As it is described in Theorem 2.3, we require $\mathcal{H}(\mathbf{X})$ to be complete in this case. To ensure that, we transform the data matrix $\mathbf{X}$ using a random representation matrix. In particular, we first generate a random representation matrix $\mathbf{H} \in \mathbb{R}^{\mathbf{d} \times \mathbf{M}}$ and then multiply it with the data matrix followed by a threshold function. Therefore, effectively, we obtain a new data matrix $\tilde{\mathbf{X}} = \sigma(\mathbf{X}\mathbf{H})$. By choosing $M$ large enough, which is $M = 1000$ in our experiments, we are able to enforce $\tilde{\mathbf{X}}$ to be full rank and thus the set of hyperplane arrangements $\mathcal{H}(\tilde{\mathbf{X}})$ is complete as assumed in Theorem 2.3. We also apply the same steps for the non-convex training, i.e., we train the networks as if we perform a two-layer network training on the data matrix $\tilde{\mathbf{X}}$. Notice that we also provide standard three-layer network training comparison, where all of three layers are trainable, in Section B.5. More importantly, after solving the convex problem (7), we need to construct a neural network that gives the same objective value. As discussed in Section 2.4, there are numerous ways to reconstruct the non-convex network weights and we particularly use **Convex-PI** and **Convex-SVM** as detailed in Section 5, which seem to have good generalization performance. We also note that these approaches are exact in the sense that they globally optimize the objective in B.3. Additionally, we have $n$ neurons in our reconstructed neural network to have fair comparison with the non-convex training with $n$ hidden neurons.

In Figure 10, we compare objective values and observe that **Convex-PI** and **Convex-SVM**, which solve the convex problem 7, obtain the globally optimal objective value whereas all trials of the non-convex training heuristics are stuck at a local minimum. Figure 11, 12, and 13 show that our convex training approaches also yield better test performance in all cases unlike the two-layer training in Section B.3. Again, for the testing phase we generate 3000 samples via the ground truth model defined above.

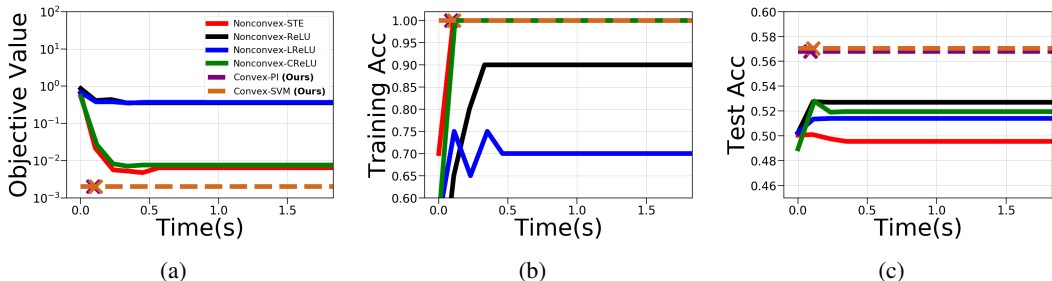

Figure 13: The setup for these figures is completely the same with Figure 11 except that we consider the $(n, d) = (20, 100)$ case in Figure 10a. This experiment also confirms better generalization of our convex training approach as in Figure 11 and 12.

### B.5 STANDARD THREE-LAYER EXPERIMENTS IN FIGURE 1 AND 2

Here, we compare our convex program in (7) with 3-layer networks trained with non-convex heuristics. Similar to the case in Section B.4 we generate a dataset via the ground truth model $\mathbf{y} = \mathrm{sgn}(\tanh(\tanh(\mathbf{XW}^{(1)})\mathbf{W}^{(2)})\mathbf{w}^{(3)})$, where $\mathbf{X} \in \mathbb{R}^{n \times d}$, $\mathbf{W}^{(1)} \in \mathbb{R}^{d \times m_1^*}$, $\mathbf{W}^{(2)} \in \mathbb{R}^{m_1^* \times m_2^*}$ and $\mathbf{w}^{(3)} \in \mathbb{R}^{m_2^*}$ and we choose $m_1^* = m_2^* = 20$ for all the experiments.

In contrast to Section B.4, we use a representation matrix $\mathbf{H} \in \mathbb{R}^{d \times M}$, where $M = 1000$, and then apply the transformation $\tilde{\mathbf{X}} = \sigma(\mathbf{XH})$ for the convex methods. We perform standard training procedure for three-layer non-convex networks, i.e., we train a fully three-layer network without any sort of representation matrix. To have fair comparison, we choose $m_1 = M$ and $m_2 = n$ for both non-convex and convex settings.

In Figure 1, we compare the objective values and observe that our convex training approaches achieve a global optimum in all cases unlike the non-convex training heuristics. We also provide the test and training accuracies for three-layer networks trained with different $(n, d)$ pairs in Figure 2. In all cases our convex approaches outperform the non-convex heuristics in terms of test accuracy.

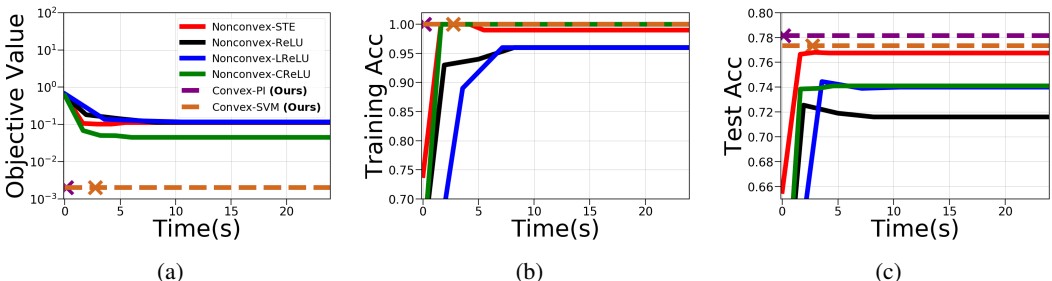

Figure 14: Performance comparison of our convex training approaches trained via (7) and non-convex training heuristics for a 10-layer threshold network training problem. Here, we use the same setup in Figure 1. As in the previous experiment, our convex training approaches yields outperforms the non-convex heuristics in both training and test metrics.

### B.6 STANDARD TEN-LAYER EXPERIMENTS IN FIGURE 14

In order to verify the effectiveness of the proposed convex formulations in training deeper networks, we consider a threshold network training problem for a 10-layer network, i.e., (10) with $L = 10$. We directly follow the same setup with Section B.5 for $(n, d, \beta) = (100, 20, 1e - 3)$. Since in this case the network is significantly deeper than the previous cases, all of the non-convex training heuristics failed to fit the training data, i.e., they couldn't achieve $100\%$ training accuracy. Therefore, for the non-convex training, we also include Batch Normalization in between hidden layers to stabilize and improve the training performance. In Figure 14, we present the results for this experiment. Here, we observe that our convex training approaches provide a globally optimal training performance and

yield higher test accuracies than all of the non-convex heuristics that are further supported by Batch Normalization.

