# OpenReview forum: "Globally Optimal Training of Neural Networks with Threshold Activation Functions"
_ICLR.cc/2023/Conference — ICLR 2023 poster_

### Official Review · Reviewer_n5sD · 2022-10-15

**Confidence:** 4
**Correctness:** 4
**Technical Novelty And Significance:** 3
**Empirical Novelty And Significance:** 3
**Recommendation:** 6

**Clarity, Quality, Novelty And Reproducibility:**

Clarity:
This paper is well-written.

Quality:
The theory looks correct. Empirical experiments were also provided to verify the proposed convex approaches.

Novelty:
As far as I know, the convex approaches to solving the training of threshold networks are novel.

Reproducibility:
Both theoretical and experimental results seem reproducible.


**Strength And Weaknesses:**

Strengths:
This paper formulated the training of weight decay regularized threshold networks as convex optimization problems. It then proposed convex approaches to optimize such neural networks, which seems appealing especially when the back-propagation fails because of the zero gradient issues. It also provides a relatively complete picture of the computational complexity of such convex approaches for different-depth networks under different data assumptions.

Weaknesses:
1. This paper only proved the polynomial-time complexity of the convex optimization problem in some special cases. In many other cases, solving this convex optimization problem can still take exponential time, which then makes the algorithm impractical. For example, when none of the layers has a width of at least $n$, the running time is exponential in the product of layer widths. This can be extremely slow since the width of each layer can easily be larger than $100$ in modern neural networks.
2. The experiments are also restricted to two or three-layer neural networks, which further reinforces my suspicion that the proposed convex approaches may not be practical for modern deep neural networks.

Some minor questions:
1. In the two-layer case when the arrangements are complete, why is the reconstruction of parameters only taking O(n)? Shouldn't it also depend on $m$ and $d$ because $W^{(1)}$ has dimension $d\times m$?
2. What's the reconstruction time when the arrangements are incomplete?

**Summary Of The Paper:**

1. This paper proved that the training of deep threshold networks with weight decay can be formulated as a convex optimization problem. The size of the convex optimization problem depends on the total number of hyperplane arrangements, which can be exponentially large in the number of samples in the worst case.
2. For two-layer threshold networks, this paper proved that the associated convex optimization problem can be solved in polynomial time when (1) the data matrix has constant rank or (2) the hyperplane arrangements of the data matrix is complete.
3. For deep threshold networks, this paper proved that the associated convex optimization problem can be solved in polynomial time when (1) the data matrix rank and the network widths are all constant or (2) one hidden layer has a width of at least the number of samples.
4. In the experiments, this paper showed that training the threshold networks using the proposed convex approaches archived better performance than the non-convex counterparts trained by back-propagation.

**Summary Of The Review:**

This paper proposed convex approaches to optimize threshold neural networks, which is particularly appealing due to the zero gradient issues in back-propagation for such networks. However, my major concern is that such convex approach can be exponentially slow in general. The experiments are also restricted to very simple networks, which further reinforces my suspicion that the proposed convex approaches may not be practical for modern deep neural networks.

---

### Official Review · Reviewer_VvcU · 2022-10-23

**Confidence:** 4
**Correctness:** 3
**Technical Novelty And Significance:** 3
**Empirical Novelty And Significance:** 2
**Recommendation:** 8

**Clarity, Quality, Novelty And Reproducibility:**

**Clarity**
- The main problem with threshold activation is that it is not continuous. The argument on gradient zero and non-zero at zero is misleading.
-  Please make the contributions more precise to avoid over-claiming. For example, the first contribution should include condition #neurons>#samples in the statement.
- When I was reading Th 2.2., I wondered how to find $d_1, \dots, d_p$ in polytime. I recommend to mention that these vectors can not be found in polytime. Indeed finding one of them is equal to optimizing threshold neural nets.
- Comment on when the completeness of arrangements are met before presenting Thm 2.3.


**Novelty**
See weaknesses *Literature review*

**Reproducibility**
See weaknesses *experiments*

**Strength And Weaknesses:**

**Strengths**
- The main strength of the paper is motivating learning threshold neural nets very well. Learning such networks is an important topic. In my opinion, the neural nets community will welcome novel algorithms and optimization techniques for neural nets with threshold activations.
- Extending memorization to deep neural nets is an interesting contribution of this paper.
- Experiments on generative and real data are a plus for this theoretical work.

**Weakness**
- *Memorization.* The paper stresses the proposed algorithm can lead to a practical algorithm. But I am not sure about this. The memorization setting (#samples<#neurons) is not standard in machine learning.
- *Presentation.* The intro and abstract may lead to a misunderstanding that threshold activation leads to convex optimization. While the memorization setting (#neurons>#samples) enables us to cast training to a convex program. I recommend explicitly declaring that the overparameterization #neurons>#samples are the key assumption enabling us to learn with a convex program.
- *Literature review* The literature on memorization is missing in this paper. The paper neglects closely related references, including [1-5] and many more references on the same topic. For example, it is very important to drive the connection between general position assumption on inputs [2] and the completeness of the arrangements in this paper.
- *Experiments.* I could not find the number of neurons, batch size, stepsize, number of epochs, and more details on experimental settings in Table 2.

**References**


1.  Rosset, Saharon, et al. "L1 regularization in infinite dimensional feature spaces." International Conference on Computational Learning Theory. Springer, Berlin, Heidelberg, 2007.
2. Bubeck, Sebastien, et al. "Network size and weights size for memorization with two-layers neural networks." arXiv preprint arXiv:2006.02855 (2020).
3. Zhang, Chiyuan, et al. "Understanding deep learning (still) requires rethinking generalization." Communications of the ACM 64.3 (2021): 107-115.
4. Pilanci, Mert, and Tolga Ergen. "Neural networks are convex regularizers: Exact polynomial-time convex optimization formulations for two-layer networks." International Conference on Machine Learning. PMLR, 2020.
5. de Dios, Jaume, and Joan Bruna. "On sparsity in overparametrised shallow relu networks." arXiv preprint arXiv:2006.10225 (2020).

**Summary Of The Paper:**

Paper develops a convex program for **memorizing** data with a two-layer neural network that extends to higher layers.  **Memorization** in neural nets is a theoretical topic that studies required conditions on a number of neurons to perfectly fit a finite number of samples. What makes this result apart from existing results in the literature is the use of the threshold activation function. The authors motivate the choice of threshold activation by (i) the compression of outputs and (ii) the biological origins of threshold activations. Despite compression and biological simulations, a network with threshold activation is difficult to optimize. There are rare algorithms to optimize networks with such activations. This paper proposes a convex program for memorization with the following guarantees:
- #neurons>#samples: the algorithm finds the weights in poly-time
- #neurons<#samples: the algorithm suffers from the complexity $O(\text{samples}^{\text{input-rank}})$

**Summary Of The Review:**

The paper presents interesting theoretical results. Yet, I believe that experimental results may lead to over-claiming and need further clarifications, and also it misses a large body of related literature.

**Post Rebuttal**
I thank the authors for their detailed response. I updated my score after reading the response.

---

### Official Review · Reviewer_tBqd · 2022-10-24

**Confidence:** 3
**Clarity, Quality, Novelty And Reproducibility:** All good.
**Correctness:** 4
**Technical Novelty And Significance:** 3
**Empirical Novelty And Significance:** 3
**Recommendation:** 6

**Strength And Weaknesses:**

Strengths:
1. The paper is well-written and easy to follow. The notations are clear.
2. Equivalenting neural network training to convex optimization under mild conditions is theoretically interesting.
3. The theoretical results seem rigorous, though I didn't check the proofs thoroughly.
4. Experiment were conducted on various datasets and the results in terms of testing accuracy and training time are encouraging.
5. The code is attached for reproducibility.

Weaknesses:
1. It seems to me that the weight decay terms in the objective function (3) are needed to derive the equivalent formulation (4). This is somewhat artificial as deep learning practitioners nowadays barely employ weight decay in training. Can the authors shed some lights on the equivalent derivation without weight decay?
2. The overall approach does not seem scale well with the depth of a neural network, unless some strong conditions are imposed. As in Theorem 3.2, the convex formulation essentially flattens the deep nested structure of the network. Therefore, it can result in exponentially many parameters to optimize, as indicated in Table 1. Please note that the point of doing neural networks is employing the deep nested structure, so as not to have exponentially many parameters, in approximating complicated functions.

**Summary Of The Paper:**

This paper formulates the problem of training neural networks with threshold activation functions to a convex optimization problem. As a result, global minima can be obtained with standard convex optimizers.

**Summary Of The Review:**

Overall, I find this paper is solid and contains interesting theoretical results. But I am concerned with its extensibility and therefore the impact, as written in Weaknesses.

---

### Decision · Program_Chairs · 2023-01-20

**Decision:**

Accept: poster

**Justification For Why Not Higher Score:**

The result is not too surprising. As the reviewers mentioned the polynomial time result requires strong conditions (one possible stronger result is if the number of parameters is larger than the number of samples, instead of number of neurons), and there is no guarantee on generalization.

**Justification For Why Not Lower Score:**

It's a reasonable theoretical result. Even though it's not surprising it's quite solid.

**Metareview: Summary, Strengths And Weaknesses:**

This paper shows that optimizing a neural network with threshold activation is equivalent to a convex program under some mild assumptions. The size of the convex program depends on the total number of hyperplane arrangements. The paper showed that for a 2-layer network, when the number of neurons is larger than the number of samples, the weights can be found in polynomial time; while in some other settings the running time can be exponential in relevant parameters. The reviewers find the result novel and interesting. There are some concerns about how the result can be generalized to deeper networks and generalization performance, but these are difficult open problems and discussions in the author response seem adequate.

**Note From Pc:**

if the above contains the word "oral" or "spotlight" please see: "oral" presentation means -> notable-top-5% and "spotlight" means -> notable-top-25%. As stated in our emails, we are disassociating presentation type from AC recommendations